# Unravelling mutational signatures with plasma circulating tumour DNA

Sebastian Hollizeck [1,2,5], Ning Wang[1,2,5], Stephen Q. Wong[1,2], Cassandra Litchfield[1], Jerick Guinto[1], Sarah Ftouni[1], Richard Rebello [3], Sehrish Kanwal [3], Ruining Dong [3], Sean Grimmond [3], Shahneen Sandhu[1,2], Linda Mileshkin [1,2], Richard W. Tothill [2,3,4], Dineika Chandrananda [1,2,6] ✉ & Sarah-Jane Dawson [1,2,3,6] ✉

The use of circulating tumour DNA (ctDNA) to profile mutational signatures represents a non-invasive opportunity for understanding cancer mutational processes. Here we present MisMatchFinder, a liquid biopsy approach for mutational signature detection using low-coverage whole-genome sequencing of ctDNA. Through analysis of 375 plasma samples across 9 cancers, we demonstrate that MisMatchFinder accurately infers single-base and doublet-base substitutions, as well as insertions and deletions to enhance the detection of ctDNA and clinically relevant mutational signatures.

Mutational signatures are distinct patterns of DNA mutations imprinted on tumour genomes by diverse processes during cancer development[1–3]. Certain intrinsic pathway errors in DNA replication or repair such as mismatch repair deficiency (dMMR) and homologous recombination deficiency (HRD) lead to mutational signatures that if correctly detected, can predict sensitivity to specific therapeutics including immunotherapy in patients with dMMR[4,5] and PARP inhibitors in patients with HRD[6,7]. Thus, mutational signature analysis is invaluable to understanding tumour biology and personalising treatments for better clinical outcomes. However, implementation of mutational signature analysis in the clinic is currently challenging due to the need for costly deep sequencing of both tumour and germline samples for accurate somatic variant calling. Moreover, tumour tissue is not always available. Alternative methods are therefore needed to help overcome these challenges.

In this work, we developed a method called MisMatchFinder which shifts away from a traditional variant-focused approach and instead utilises read-based somatic variant inference prior to mutational signature deconvolution (Fig. 1a). The tool is suitable for low-coverage whole-genome sequencing (LCWGS) data (<10×) and is tailored for circulating tumour DNA (ctDNA), thus offering a means to characterise mutational signatures without the need for invasive biopsies.

## Results

### Description and performance of MisMatchFinder in simulated and clinical sequencing data

The MisMatchFinder algorithm identifies mismatches within reads compared to the reference genome and filters background noise unrelated to somatic mutations through (i) the use of high thresholds for mapping and base quality, (ii) strict consensus between overlapping read-pairs, (iii) gnomAD-based germline variant filtering[8], and (iv) a ctDNA-centric fragmentomics filter[9] (Fig. 1a). A read depth filter is optional and allows for scaling beyond low sequencing coverage, providing additional flexibility. Once high-confidence mismatches are selected, these can be used to extract novel signatures or fit pre-defined ones[2]. MisMatchFinder is equipped to analyse single-base, and doublet-base substitutions, as well as insertions and deletions (indels). In this study, we utilised all three types to assign weights to the mutational signatures in the Catalogue Of Somatic Mutations In Cancer (COSMIC) database (version 3.2)[1] using non-negative matrix factorization with quadratic programing[10]. To identify signatures that were over-represented in ctDNA rather than normal cell-free DNA (cfDNA), we derived detection thresholds using signature weights from a panel of healthy cfDNA controls (details in "Methods").

To optimise the performance of MisMatchFinder, we systematically tested each filtering step using both in silico and patient

---

[1]Peter MacCallum Cancer Centre, Melbourne, VIC, Australia. [2]Sir Peter MacCallum Department of Oncology, The University of Melbourne, Melbourne, VIC, Australia. [3]Centre for Cancer Research, The University of Melbourne, Melbourne, VIC, Australia. [4]Department of Clinical Pathology, The University of Melbourne, Melbourne, VIC, Australia. [5]These authors contributed equally: Sebastian Hollizeck, Ning Wang. [6]These authors jointly supervised this work: Dineika Chandrananda, Sarah-Jane Dawson. ✉e-mail: Dineika.Chandrananda@petermac.org; Sarah-Jane.Dawson@petermac.org

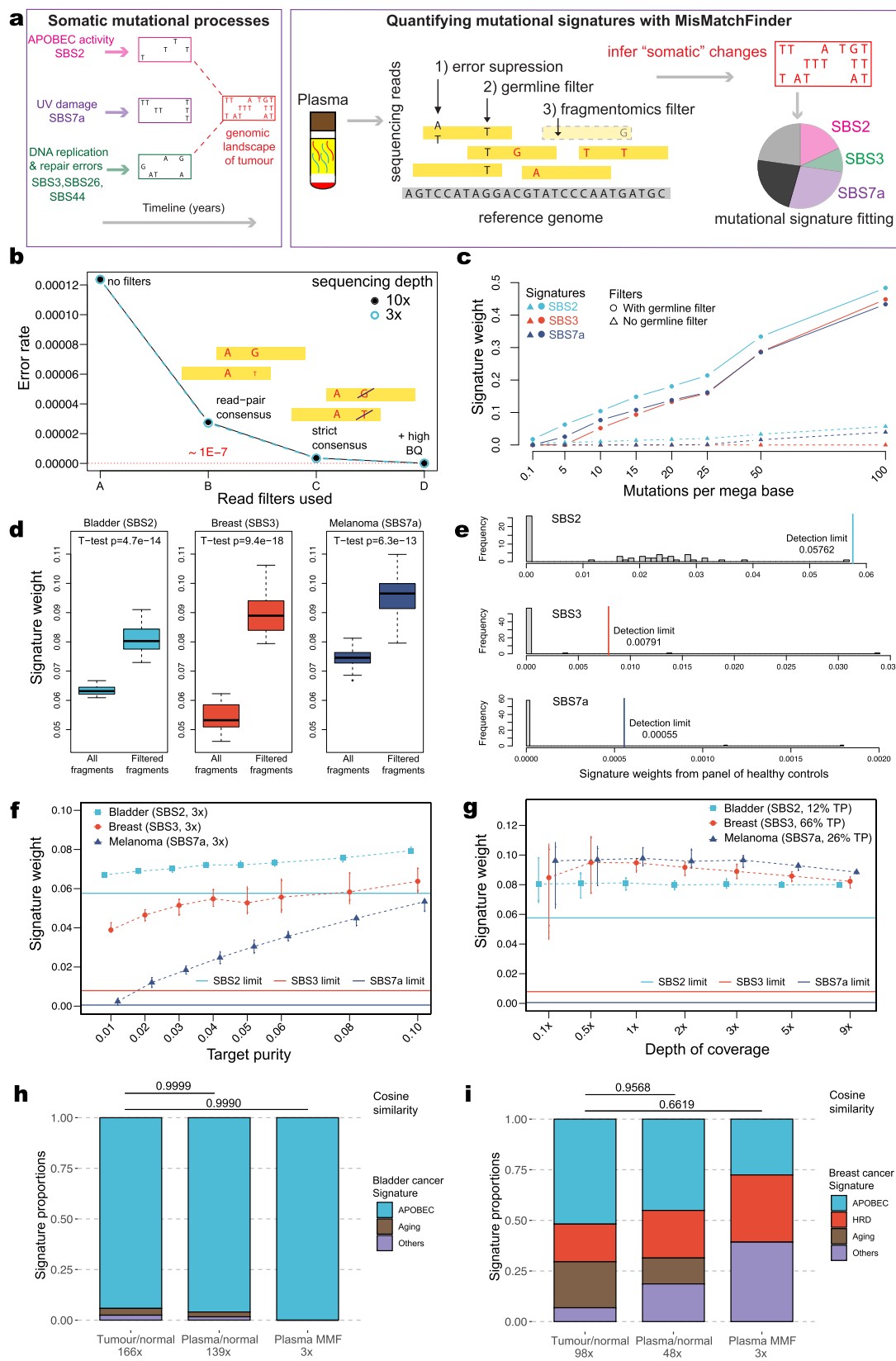

data. We first simulated "clean" reads without germline or somatic variants where any deviations from the reference genome would solely represent sequencing errors. We determined that at both 3× and 10× depth, when only selecting for mismatches where paired-reads had strict consensus and high base qualities (BQ ≥ 32), only 1 error in 10 million bases was miscounted as a variant (Fig. 1b). This

error rate is far lower than most tumour mutational burden estimates[11].

We next investigated the effectiveness of the germline filter by separately spiking in somatic variants related to multiple different signatures into LCWGS data (3×) from a healthy cfDNA control. For this exercise, we selected two "peak-like" signatures of SBS2 (associated

**Fig. 1 | Description and performance of MisMatchFinder in simulated and clinical sequencing data. a** Schematic describing the MisMatchFinder algorithm for within-sample mutational signature detection in a liquid biopsy context. **b** Sequencing error rates following distinct filtering approaches applied to LCWGS data simulated to only contain sequencing errors. Error rates are shown after (A) no filters are applied; retaining all mismatches. The following filters are incremental i.e. (C) is a subset of (B) and (D) a subset of (C). (B) Read-pair consensus; retains only mismatches within paired-read overlaps after building consensus for differing base and/or quality, (C) Strict consensus; only retains mismatches that have the same base between paired-reads, and (D) +High BQ; retains mismatches with the same base in both reads with base quality (BQ) ≥ 32. Data are provided in Zenodo [https://doi.org/10.5281/zenodo.13845728]. **c** Effect of gnomAD germline variant filtering on signature detection. Assessed for the APOBEC signature SBS2, the HRD signature SBS3 and the UV-damage signature SBS7a in LCWGS data simulated with varying mutational burdens with depth of coverage fixed at 3×. Source data are provided as a Source Data file. **d** Effect of the fragmentomics filter to enrich for mismatches originating from ctDNA. Signature weights presented for SBS2, SBS3, and SBS7a, derived from bladder cancer (estimated at 12% tumour purity (TP)), *BRCA1*-mutant breast cancer (66% TP), and melanoma (26% TP) plasma datasets from three cancer patients. For each cancer type, boxplots represent 20 in silico replicates at 3× sequencing depth. Box plots indicate median (middle line), 25th–75th percentile (box) and 1.5 times the inter-quartile range from the first and third quartiles (whiskers). Outliers were omitted. Two-sided *t*-tests were performed to compare signature weights using all fragments and using filtered fragments. Source data are provided as a Source Data file. **e** The distributions of signature weights and detection thresholds (vertical lines) for SBS2, SBS3, and SBS7a from 60 healthy control plasma LCWGS datasets. Source data are provided as a Source Data file. **f** Effect of tumour purity (TP) in plasma on the limit of signature detection applying all filters and detection limits described in **b**–**e** for depth of coverage fixed at 3×. Assessed for signatures SBS2, SBS3 and SBS7a in 20 ctDNA-healthy admixtures from a bladder cancer, a *BRCA*-mutant breast cancer and a melanoma patient,

respectively. For each cancer type, each vertical line represents a boxplot of the signature weights of 20 in silico replicates at different TPs. The bounds of each vertical line are the 25th to 75th percentile and the median signature weights are denoted by the symbols. The horizontal lines denote the detection thresholds per signature type derived from healthy plasma controls. Source data are provided as a Source Data file. **g** Impact of sequencing coverage on the limit of signature detection applying all filters and detection limits described in **b**–**e** for TP at the original levels. Assessed for SBS2, SBS3 and SBS7a in 20 ctDNA-healthy admixtures from bladder cancer, a *BRCA1*-mutant breast cancer and a melanoma patient, respectively. For each cancer type, each vertical line represents a boxplot of the signature weights of 20 in silico replicates at different depths of coverage. The bounds of each vertical line are the 25th to 75th percentile and the median signature weights are denoted by the symbols. The horizontal lines denote the detection thresholds per signature type derived from healthy plasma controls. Source data are provided as a Source Data file. **h** Comparison of grouped SBS mutational signatures detected from paired tumour tissue and plasma in a bladder cancer patient. The three columns of signatures were obtained from somatic variants called using 1) high-coverage paired tumour-germline data, 2) plasma-germline data, and 3) variants inferred from low-coverage plasma data (3×) without a germline control using MisMatchFinder (MMF). The signatures assessed were those which have previously been found in bladder cancers[1] (APOBEC: SBS2, SBS13; Aging: SBS1, SBS5; and Others: SBS8, SBS29, and SBS40). Pairwise cosine similarities of signature sets from 2) and 3) against the tumour-germline signatures are annotated above the plots. Source data are provided as a Source Data file. **i** Comparison of grouped SBS mutational signatures detected from paired tumour tissue and plasma in a *BRCA1*-mutant breast cancer patient. The three columns represent the same groups as for **h** The signatures assessed were those which have been previously found in breast cancers[1] (APOBEC: SBS2, SBS13; HRD: SBS3; Aging: SBS1, SBS5; Others: SBS8, SBS9, SBS17a, SBS17b, SBS18, SBS37, SBS40, and SBS41). Source data are provided as a Source Data file.

with Apolipoprotein B mRNA editing enzyme catalytic polypeptide-like; APOBEC) and SBS7a (related to Ultraviolet; UV-damage); both very distinct and dominant in C > T mutations. In contrast, we also investigated the "flat", non-distinctive SBS3 HRD signature, which is more difficult to fit, to assess the germline filter. We observed how removing variants annotated within gnomAD enhanced the detection of all three signatures, indicating this was a stringent but necessary filtration step (Fig. 1c). Further testing of this filter using additional signatures (SBS4, SBS7c, SBS13, and SBS44) across mutational loads simulated according to the COSMIC database, highlighted the generalisability of the germline filter effect (Supplementary Fig. 1).

We then assessed a fragmentomics filter which selects for paired-reads in specific size ranges previously evidenced to be enriched for ctDNA in plasma[9] (Supplementary Fig. 2). We applied MisMatchFinder to plasma LCWGS at 3× from bladder cancer (tumour purity (TP) 12%), breast cancer (TP 66%), and melanoma (TP 26%) patients with SBS2, SBS3, and SBS7a signatures, respectively and compared signature weights before and after the filter. This was assessed across 20 in silico replicates in each cancer type. We observed statistically significant increases (*p*-value < $10^{-10}$) in all signatures investigated, with a median 1.27-fold increase for SBS2, 1.66-fold increase for SBS3 and 1.28-fold increase for SBS7a, respectively (Fig. 1d). This established that the fragmentomics filter enhanced the tumour signal in plasma LCWGS to facilitate signature detection.

Using the same patient data as above, we then assessed the combined effect of all previously described filters by quantifying the sensitivity of MisMatchFinder to detect specific mutational signatures as a function of changing TP and sequencing depth. To understand the limits of detection, we first analysed signature weights from MisMatchFinder across a panel of 60 healthy plasma LCWGS controls. A beta distribution was fitted to the healthy control weights per mutational signature and the 99th quantile was selected as the detection threshold to identify signatures that were over-represented in ctDNA (Fig. 1e and 'Methods'). Thereafter, when applying MisMatchFinder to

the plasma LCWGS data from the bladder cancer, breast cancer, and melanoma patients, we observed that signature detection increased with increasing TP from 1% to 10% across SBS2, SBS3 and SBS7a (Fig. 1f) when sequencing depth was fixed at 3×. When varying the sequencing depth but fixing TP at the original values, all three signatures were identifiable even at 0.5× (Fig. 1g). We further explored additional coverage and purity combinations for signature detection and noted that coverage ≤1× introduced substantial variability, particularly for the flat SBS3 signature which was more apparent in samples with low TP (<10%) (Supplementary Figs. 3 and 4), suggesting that higher sequencing depths should be used in scenarios with low TP.

As further orthogonal validation, we directly compared mutational signature detection using (i) gold-standard high-coverage tumour-germline paired variant calling then (ii) paired variant calling using WGS of plasma and germline samples and finally (iii) MisMatchFinder for LCWGS of plasma (3×) without the matched germline sample, in a patient with bladder cancer and another patient with *BRCA1*-mutant breast cancer. Here, MisMatchFinder was found to reliably detect clinically relevant signatures identified in the tumour, including APOBEC and HRD signatures, as well as other signatures which have previously been found in bladder or breast cancers, respectively[1]. The cosine similarities between signatures from MisMatchFinder in 3× plasma datasets and high-coverage, tumour-germline somatic calling were 0.999 for bladder cancer and 0.662 for breast cancer patients (Fig. 1h, i). Furthermore, we performed this tumour/plasma concordance analysis across varying sequencing depths (9× down to 0.1×) and tumour purities (10% down to 1%). These results showed that the key mutational signatures remain consistent even at extreme values of coverage and TP (Supplementary Fig. 5).

## MisMatchFinder signature analysis for detecting cancer-related and clinically relevant signatures

After evaluating MisMatchFinder's performance, we appraised its capacity to extract different mutational signatures between plasma

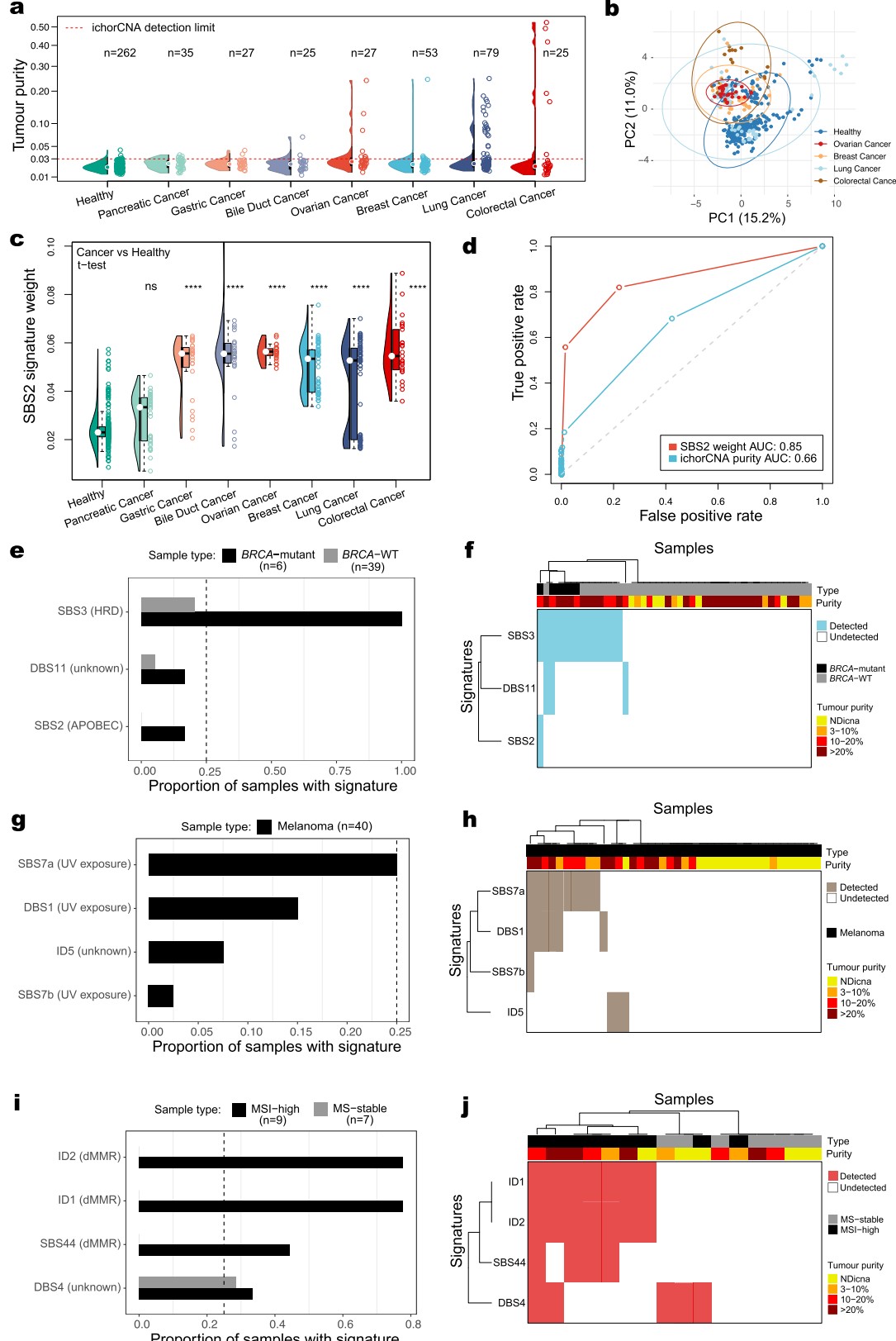

samples from healthy and cancer patients. We first utilised a publicly available[12] pan-cancer plasma LCWGS dataset (1–2×), encompassing 271 samples with low TP across 7 cancer types and 262 healthy controls (Fig. 2a). PCA analysis using all SBS signatures (apart from those related to sequencing artefacts) showed moderate separation between cancer and healthy samples (Fig. 2b). Further analysis showed that the

APOBEC-linked SBS2 signature can be used as a major discriminatory signature, separating cancer samples from healthy samples, across most cancer types (Fig. 2c). Furthermore, we identified that Mis-MatchFinder's quantification of the SBS2 signature improved ctDNA detection over copy number-based tumour purity assessment in this ultra-low-coverage pan-cancer cohort (Fig. 2d). These findings are

**Fig. 2 | MisMatchFinder signature analysis for detecting cancer-related and clinically relevant signatures. a** Tumour purity distribution in a pan-cancer plasma cohort comprising 271 samples across 7 cancer types and 262 healthy controls using 1-2x LCWGS. The red dashed line denotes ichorCNA's detection limit for tumour purity estimation. Source data are provided as Supplementary Data 3. **b** Principal component analysis of healthy controls against four cancer types with high estimated tumour purity analysed using all SBS signatures excluding those linked to sequencing and library preparation artefacts. Source data are provided as Supplementary Data 3. **c** Distribution of the APOBEC-enzyme activity linked SBS2 signature weights across the cohort in **a**. The number of asterisks quantifies the statistically significant difference between each cancer type with the healthy control group using a two-sided *t*-test (*$p \leq 0.05$, **$p \leq 0.01$, ***$p \leq 0.001$, ****$p \leq 0.0001$, ns non-significant). The exact *p*-values from left to right are: 1, 3.3e-20, 6.9e-31, 5.5e-29, 1.1e-32, 8.1e-24, and 2.0e-28. Box plots indicate median (middle line), 25th to 75th percentile (box) and 1.5 times the inter-quartile range from the first and third quartiles (whiskers). Outliers were omitted. Source data are provided as

Supplementary Data 3. **d** Receiver Operating Characteristic (ROC) curves and area under the curve (AUC) values for discriminating cancer from healthy plasma using tumour purity estimates from **a** and SBS2 weights from (**c**). Source data are provided as Supplementary Data 3. **e** Detection frequency of the top SBS/DBS and indel-based signatures previously found in this cancer type, for 45 breast cancer patients with known *BRCA1/2* mutational status. Source data are provided as Supplementary Data 1. **f** Hierarchical clustering of all samples and signatures in **e** annotated by their estimated tumour purity in plasma. NDicna relates to samples with TP below the limit of detection of ichorCNA. Source data are provided as Supplementary Data 1. **g** As in **e** for 40 patients with melanoma with varying TP. Source data are provided as Supplementary Data 1. **h** As in **f** for cohort and signatures in **g**. Source data are provided as Supplementary Data 1. **i** As in **e** for 16 patients with colorectal cancer with known microsatellite instability (MSI) status. Source data are provided as Supplementary Data 2. **j** As in **f** for cohort and signatures in **i**. Source data are provided as Supplementary Data 2.

consistent with the APOBEC signature being one of the most widely detected signatures in human cancers[1] which could be utilised to enhance cancer detection through liquid biopsy.

Next, we analysed three independent clinical cohorts encompassing 101 patients, to further validate MisMatchFinder's capability to identify distinct clinically relevant signatures. We first analysed a cohort of 45 breast cancer patients with known *BRCA* mutation status and as expected, identified strong enrichment of the HRD SBS3 signature (Fig. 2e, f). In this cohort, we compared the performance of MisMatchFinder to shallowHRD[13], a computational tool for detecting HRD from low-coverage tumour sequencing data. MisMatchFinder detected SBS3 in all 6 cases (100%) with known *BRCA1* or *BRCA2* germline mutations, compared to shallowHRD which only predicted HRD in 5/6 (83%) of these cases (Supplementary Fig. 6). MisMatchFinder also detected an SBS3 signature in 8/39 patients without a germline *BRCA* mutation, and 6 out of these 8 cases were also validated by shallowHRD (Supplementary Fig. 6). In parallel, we also looked at deletions within microhomology regions for the 6 *BRCA* mutation cases and the 39 *BRCA* wild-type cases. As expected, we observed that the cases with a known *BRCA* germline mutation had the highest number of microhomology deletions (Supplementary Fig. 7). Importantly, the 8 *BRCA* wild-type cases which showed an elevated SBS3 mutational signature profile, also had an elevated number of microhomology deletions compared to *BRCA* wild type cases with no SBS3 detected (Mann–Whitney *U*-test, *p*-value = 0.023), adding further evidence for the presence of HRD in these cases (Supplementary Fig. 7). In a cohort of 40 melanoma patients, the UV exposure signatures SBS7a, DBS1, and SBS7b were among the top signatures detected among all cases, suggesting that these UV-damage signatures may enhance ctDNA detection in this disease (Fig. 2g, h). Lastly, in a cohort of 16 colorectal cancer patients with known microsatellite instability (MSI) status, detection of dMMR indel-based signatures (ID1, ID2) and SBS44 was confined to the MSI-high cases showing specificity of the MisMatchFinder tool beyond SBS signatures to encompass indel signature detection for improved dMMR assessment (Fig. 2i, j).

## Discussion

Our study demonstrates the feasibility of cost-effective ctDNA analysis from LCWGS data for clinically relevant mutational signature detection that can be applied to single samples. We recommend that at least 1x coverage for plasma sequencing data is essential for detecting mutational signatures with distinct profiles such as SBS2 and SBS7 even for estimated tumour purity levels >3%. However, for "flat", non-distinctive signatures such as SBS3, higher sequencing depth (>3×) is necessary to ensure robust results, particularly in samples with <10% TP. While these are conservative recommendations for recovering mutational signatures in liquid biopsy samples based on our analyses, the limits of detection will also be influenced by the variable

mutational burden across different tumour types. Whenever possible, we recommend using a panel of healthy plasma controls of similar sequencing depth and other pre-analytical factors to derive detection thresholds for a high-confidence ctDNA-based mutational signature signal.

Despite the advances that MisMatchFinder brings to the field, our current study has some limitations. Whilst we have undertaken a detailed assessment of the performance of our tool for SBS signature detection, in silico simulation of indels to assess the detection of different ID signatures is more challenging and has not been performed. Indels do not emerge at random positions within the genome and are enriched in certain sequence contexts such as short tandem repeats and microhomologies[14,15]. This has limited our ability to assess ID signature detection using MisMatchFinder with simulated data. Furthermore, whilst MisMatchFinder is able to retrieve somatic variants from LCWGS plasma sequencing data, it still relies on established tools for signature fitting which have been developed for tumour tissue sequencing data and can vary in performance[16,17]. In the future, with the generation of more matched tumour/normal/plasma sequencing datasets across various cancer types, signature fitting algorithms could be specifically tailored for ctDNA and may further improve mutational signature assessment in liquid biopsies.

Other studies have explored the use of machine learning to identify mutational patterns from ctDNA sequencing analysis to enhance cancer detection[18,19]. Whilst the incorporation of machine learning approaches can improve the sensitivity of ctDNA detection, they rely on training and validation across large datasets, and are also susceptible to batch effects[20] which limit their ability to be easily translated for widespread clinical adoption. In contrast, MisMatchFinder can facilitate broader integration of mutational signature analysis from serial blood collections, to enable the evolution of mutational processes to be monitored over time, particularly in patients receiving therapy. MisMatchFinder holds the potential for novel insights into the predictive and prognostic roles of mutational signatures in clinical settings and may provide opportunities to inform clinical decisions through optimised personalised cancer treatment strategies.

## Methods

### Ethics information for plasma samples and sequencing data generation

This study utilised a combination of sequencing datasets downloaded from public repositories and those generated at the Peter MacCallum Cancer Centre (PMCC) and the University of Melbourne. Forty-six patients with breast cancer, 41 melanoma patients and 60 healthy controls were recruited following informed consent with each study approved by the PMCC Human Research Ethics Committee (Breast HREC 15/72; Melanoma HREC 11/105 and 07/38; Healthy controls HREC

17/56). In addition, one patient (case 1205) with bladder cancer, originally diagnosed as cancer of unknown primary, was recruited following informed consent within the SUPER cohort study at PMCC (HREC 13/62 and 11/117). All patient consent was provided in written/signed form. Sex and gender were not considered in the study design as this information was not necessary for the particulars of the study.

Libraries for low coverage whole genome (LCWGS) sequencing from plasma cfDNA were generated at the PMCC Genomics Core Facility using the NEBNext® Ultra™ II DNA Library Prep Kit following the standard protocol and sequenced on the Illumina Novaseq platform to generate 100 bp paired-end reads at an average sequencing depth of ~9× (Supplementary Data 1).

A LCWGS dataset (~0.3–4×) of plasma cfDNA from 16 colorectal cancer patients with known microsatellite instability status, as well as 21 healthy individuals, was accessed from the European Genome-phenome Archive (EGA) (accession number EGAS00001006377, Supplementary Data 2). A pan-cancer LCWGS dataset of plasma cfDNA was accessed from EGA (accession number: EGAS00001003611) containing 271 samples across 7 cancer types and 262 healthy cfDNA controls with average coverage between 1× and 2× (Supplementary Data 3).

Paired-end sequencing reads from all low-coverage plasma datasets were aligned to the human reference genome (GRCh38) using bwa-mem[21] (version 0.7.17) with alternate-contig aware mapping. Presumed PCR and optical duplicates were marked using the Picard tools software suite (version 2.17.3).

## The MisMatchFinder algorithm

MisMatchFinder identifies "mismatches" from the reference genome with a mismatch in this work considered as any position in an aligned read which does not show the same base as the reference genome at the aligned position. The mismatch inherits all the metrics of the read, such as mapping quality, base quality and read position.

Within MisMatchFinder, the "MD" and "CIGAR" tags of sequencing reads were used to reconstruct the sequence of the read and its positions where the read showed a different base than the reference. MisMatchFinder allows several filtering criteria to reduce this set of mismatch sites to decrease the impact of germline variants, and sequencing errors and to increase the probability of retaining somatic variants.

MisMatchFinder allows for user-defined thresholds for standard filters such as mapping quality (MQ) and base quality (BQ). The default cut-offs are $MQ = 20$, and $BQ = 65$ with the BQ reflecting the sum of base qualities of paired reads in regions of overlap. There are further filters applied to set the minimum average base quality across a read or read pair (default: 25) and the minimum and maximum number of mismatches per read (default: 1 and 15, respectively). Users may specify a minimum and maximum length of a fragment for paired-end sequencing and this filter was used to enrich for ctDNA reads in this study. Finally, MisMatchFinder filters out alignments flagged as secondary or presumed PCR and optical duplicates.

Users can provide a whitelist bed-file and in this work, we have used this option to restrict the analysis to only highly mappable regions of the genome (unique mappability ≥ 85%). A blacklist is also optional but was not used in this study.

MisMatchFinder allows several options to calculate internal consensus in regions where paired reads overlap to adjust for differences between forward and reverse reads. In many variant calling methods, these differences are used by measuring the "strand bias" or "strand balance probability" by looking at a specific locus and evaluating the discrepancy of all forward and reverse reads at that position. As our method examined each read/fragment independently, these bias estimates cannot be calculated, however, in the overlapping region of both reads, a consensus can be generated. If both reads agreed on the mismatch, the BQ of the reads were summed to emphasize the increased evidence for these variants. In contrast, if they disagreed the

base with the higher quality was used and its quality was decreased by half of the BQ of the lower quality base. To increase the stringency of the method, MisMatchFinder can be configured to only use mismatches in regions where paired reads overlap ('–onlyOverlap'), which significantly reduces the number of sequencing errors which were retained in the final analysis. For the most stringent analysis, the user can also enable the '–strictOverlap' option in addition to '–onlyOverlap'. Here, for mismatches found in regions where paired reads overlap, the tool will only consider a mismatch if both reads agree with each other.

Users can provide a file containing any variants they wish to be excluded from the analysis and MisMatchfinder uses the echtvar (https://github.com/brentp/echtvar) encoded file format for near-instant variant lookup. In this study, we used the gnomad v3.1.2 file distributed with echtvar to filter for variants found in this database.

For compatibility with downstream signature deconstruction methods, MisMatchFinder outputs a VCF for each sample analysed in concordance with the v4.2 VCF specifications. For further filtering, MisMatchFinder also reports the number of independent fragments that support the mismatch as the 'MULTI' field in the INFO column, to allow the user to apply further confidence thresholds if higher-depth sequencing is available.

The MisMatchFinder tool is programmed in Rust. We offer MisMatchFinder as a fully compiled static binary file, as well as the source codes and libraries used within the tool in: https://atlassian.petermac.org.au/bitbucket/projects/SJDAW/repos/mismatchfinder/browse.

Other data required for filters such as gnomAD variants for the germline filter, and high mappability regions as white-listed regions used in this study can be found in the Zenodo repository (https://doi.org/10.5281/zenodo.13845728)[22].

## Signature fitting

Signature fitting was performed with the sigminer[10] R package v2.3.0 with default settings using the Cosmic v3.2 human reference genome GRCh38 signature catalogue for SBS, DBS, and ID signatures. Sigminer also corrects for potential GC bias of sequencing through background estimation of variants. In this study, signatures associated with sequencing and library preparation artefacts were removed and the weights renormalised, so that the sum of weights for each sample equalled 1.

## Signature detection and usage of a panel of healthy plasma controls

Signature detection thresholds were computed by fitting a beta distribution for each signature across healthy samples from the same source, after removing the two samples with the highest and lowest weights, to ensure a more conservative fit and lessen the effect of potential outliers. For a more stable fitting of signatures, the R package fitdistrplus[23] (v 1.1-8) was used with "moment matching estimation" and the mean and the standard deviation of the controls were used as initial states for the optimisation. The 99th quantile was used as the detection threshold for all signatures.

## Comparison of MisMatchFinder plasma-based analysis with matched tumour tissue data

Temporally matched tumour (fresh frozen biopsies), blood germline and plasma samples were collected from a bladder cancer patient (sample id: 1205) and a *BRCA1*-mutant breast cancer patient (sample id: MBCB196) to directly compare signature analysis using gold-standard tumour-germline paired variant calling with MisMatchFinder. For this purpose, somatic variant calling was performed with the DRAGEN pipeline[24] (v4.2.4) using the human reference genome GRCh38, on all canonical chromosomes according to the user guide. The output VCF files from DRAGEN were further processed through an in-house post-processing workflow to prioritise small somatic variant calls (https://

github.com/umccr/umccrise/tree/master). Both SNV and indel results were used for the reconstruction of signatures. The deletions within microhomology regions were labelled by VarSCAT[15] (v1.1.0), where the definition of deletion within microhomology was based on HRDetect[25,26].

To comprehensively assess the concordance between signatures captured in plasma compared to those detected in the tumour, somatic variants were called from high-coverage, tumour-germline sequencing data of each patient with the DRAGEN pipeline (v4.2.4) with downstream filtering as described above. The resulting somatic variants were fitted to mutational signatures (COSMIC v3.2) with sigminer (v2.3.0). Then the same workflow was used to call somatic variants and signatures from the high-coverage, plasma-germline paired data. The signatures of each sample were grouped into similar categories where possible (e.g. SBS2 and SBS13 grouped as APOBEC) and then intersected with signatures previously found in bladder and breast cancers, respectively[1] to assess cancer-type specific signature concordance. We used cosine similarities in these signature subsets as compared to tumour-germline results to quantify signature concordance in all associated analyses.

### Comparison of MisMatchFinder and shallowHRD for HRD signature detection

shallowHRD[13] (v1.13, hg38 version, default parameters) was run on the plasma LCWGS from 45 breast cancer patients with known *BRCA* mutant status and compared directly with MisMatchFinder. QDNAseq[27] (v1.32.0) was used to generate genome-wide copy number values using 50 kb windows as input for shallowHRD.

### Mutation assays

Mutations in *BRCA1* and *BRCA2* genes for the 45 breast cancer patients were detected through routine clinical sequencing performed at PMCC Molecular Pathology as part of standard clinical care.

### Inferring tumour purity from LCWGS of plasma

Tumour purity was estimated in each LCWGS dataset using ichorCNA[28] (version 0.2.0) implementing a window size of 1 Mb (--window) and restricting counts to autosomes (--chromosome), using default values for the other parameters.

### Simulating "clean" sequencing data

Simulation with 'art_illumina'[29] (v2.5.8) was performed with a read length of 100 bp, a mean insert size of 166, and an insert size standard deviation of 24 using the profile of the HiSeqX TruSeq, v2.5 (HSXt). These metrics were computed from the in-house healthy plasma sequencing data to more closely represent cfDNA characteristics. The fastq files produced this way were then aligned and processed as described previously.

### Spike-in for germline filter and mutational burden estimation

For all the assessed signatures, variants were selected from the COSMIC somatic variant catalogue (v3.2) to replicate each respective signature profile. Each variant was annotated with its trinucleotide context in the GRCh38 reference genome and normalised to only contain pyrimidine nucleotides at the centre. The reverse complement was used when the central nucleotide was a purine. The number of variants required to be spiked into a healthy plasma control was calculated using the target mutation rate $r_m$, and the genomic length $n_{genome}$ of the reference and normalised by $1 \cdot 10^6$ to equate to mutations per million.

$$\frac{dx}{dy}n(vars) = \frac{r_m}{1 \cdot 10^6} \cdot n_{genome} \quad (1)$$

This formula equates to about 300 variants at $r_m = 0.1$ up to 300,000 at $r_m = 100$. The bamsurgeon tool (v1.2.1) was used to spike in these variants at a constant allele frequency of 0.1. To allow bamsurgeon to make little to no changes, as expected in LCWGS, we changed the minimum mutation reads (-minmutreads) to 0, the minimum depth to 1 (-mindepth) and allowed a higher-than-normal skew in the coverage difference before and after (-d 0.7). Lastly, a fixed seed was used to create reproducible results. The spiked-in variants were then analysed with sigminer to verify that the signatures were conserved.

### Limit of detection simulations for tumour purity (TP) and sequencing depth

We established the limit of detection as a function of both coverage depth and tumour purity for SBS2 (bladder cancer), SBS3 (breast cancer), and SBS7a (melanoma) using LCWGS of patient plasma samples (sample ids: 1205, MBCB196 and PMC1141_18_02_2019, respectively). These samples are referred to as the "source data" below. The starting tumour purities were estimated at 12% TP for the bladder cancer case, 26% TP for the melanoma case and 66% TP and 11% TP for the two breast cancer cases.

### Sequencing depth analysis

Each source data sample was downsampled with fixed seeds at each downsampling step. Downsampling was performed with samtools v1.13[30] using a subsampling fraction ($f$) based on the number and length of reads of the source data ($n_{reads} \cdot len_{read}$), the required number of reads to reach the desired depth of coverage (depth$_t$), and the size of the reference genome in base pairs ($n_{genome}$):

$$f = \frac{n_{genome} \cdot depth_t}{len_{read} \cdot n_{reads}} \quad (2)$$

To ensure a representative result, each source dataset was downsampled 20 times per target depth.

### Purity-based limit of detection

As in the depth-based downsampling, the seeds for different target purities were kept stable to ensure the same selection of reads was used at the different levels. To reach the target tumour purity, but remain at a stable depth, reads from the source data were removed and replaced with reads from a healthy control sample. The fraction of reads taken from the source data ($f_{tumor}$) is calculated from the original tumour purity of the source data ($p_s$) and the target tumour purity ($p_t$)

$$f_{tumour} = \frac{p_t}{p_s} \quad (3)$$

The subsampling fraction for the healthy reads was calculated using the number of reads from the source data ($n_s$) and the number of reads from the donor data ($n_d$).

$$f_{normal} = (1 - f_{tumour}) \cdot \frac{n_s}{n_d} \quad (4)$$

A random sampling of both the source and the healthy data was performed with samtools before merging into in silico datasets at various target purities (1–10%). In order to have an accurate representation of the sampling distribution, we performed this replacement 20 times per target purity in each source dataset.

### Other statistical analysis

Principal Component Analysis was carried out using the prcomp R function (version 4.2.0), allowing for scaling of signature weights. Centering was omitted, to maintain the information that signature weights cannot be negative. Cosine similarities between tumour/plasma signature sets were measured with the cosine function within the lsa R package (version 0.73.3).

## Reporting summary

Further information on research design is available in the Nature Portfolio Reporting Summary linked to this article.

## Data availability

The pan-cancer LCWGS dataset used in this study is available from the European Genome-phenome Archive (EGA) under accession number: EGAS00001003611. The signature weights for this cohort are contained within Supplementary Data 3. The sequencing data for the 16 colorectal cancer patients with known microsatellite instability status, as well as 21 healthy individuals, is available from EGA under accession number EGAS00001006377. The signature detection information for this cohort is contained within Supplementary Data 2. The sequencing data from breast cancer and melanoma patients, as well as the in-house healthy controls, are available under EGA accession numbers EGAS00001007593 and EGAS50000000569. The signature detection information for these cancer samples is contained within Supplementary Data 1. The tissue, germline and plasma sequencing for the bladder cancer case is available from EGA under accession number EGAS50000000452. Access will be granted by application to the relevant Data Access Committees for each cohort, and will be governed by the provisions laid out in the associated informed consent for each cohort and the terms contained in the relevant Data Access Agreements. Other data required for filters such as gnomAD variants for the germline filter, and high mappability regions as white-listed regions used in this study can be found in the following Zenodo repository: https://doi.org/10.5281/zenodo.13845728[22]. Source data are provided with this paper.

## Code availability

We offer MisMatchFinder as a fully compiled static binary file, as well as the source codes and libraries used within the tool in the following repository: https://atlassian.petermac.org.au/bitbucket/projects/SJDAW/repos/mismatchfinder/browse. The reference files and small demo files for MisMatchFinder can be found in Zenodo (https://doi.org/10.5281/zenodo.13845728)[22].

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

## Acknowledgements

The authors thank the patients whose biospecimens were used as part of this research. Funding support for this research was provided by two National Health and Medical Research Council (NHMRC) grants (#1196755 for S.-J.D. and #2019655 for D.C). We thank the Molecular Genomics Core at the Peter MacCallum Cancer Centre and the Personalised Genomic Pathology Program at the University of Melbourne Center for Cancer Research for assistance with sequencing.

## Author contributions

Conception and design: S.H., N.W., S.W., D.C., and S.-J.D. Acquisition of data: S.H., N.W., S.W., C.L., J.G., S.F., R.R., S.K., R.D., S.G., S.S., L.M., R.T., D.C., and S.-J.D. Analysis and interpretation of data: S.H., N.W., D.C., and SJ.D. Writing, review and /or revision of the manuscript: S.H., N.W., S.W., D.C., and S.J.D. Study supervision: D.C. and SJ.D. All authors approved the final version of the manuscript.

## Competing interests

The authors declare the following competing interests: S.-J.D. is an advisory board member for Adela. The remaining authors declare no competing interests.
