## [Peer Review File · Nature Communications]

Unravelling mutational signatures with plasma circulating tumour DNA

Corresponding Author: Professor Sarah-Jane Dawson

Version 0:

Reviewer comments:

Reviewer #1

(Remarks to the Author)

Hollizeck et al present a compelling study demonstrating that mutational signatures can be inferred successfully from plasma circulating tumour DNA. They introduce a new tool MisMatchFinder that can call SNVs and indels from low coverage ctDNA sequencing and subsequently use established methods in the field (implemented in sigminer) to infer the mutational signatures in a variety of plasma ctDNA cohorts. The authors have paid particular attention to filtering mutation calls so as to ensure the optimal capture of cancer-related signal, and demonstrate this convincingly in the first part of the manuscript. The statistics are sound. I believe this is an important study opening up exciting possibilities in exploring mutational signatures for the early detection of cancer in liquid biopsies, and the authors provide some initial evidence this could be useful particularly for MSI high CRCs and HR deficient breast cancers. I have a few questions/comments that I believe would strengthen the study:

1. The authors tested the effectiveness of their filters by spiking somatic variants linked to the SBS7a UV damage signature and SBS3 HRD signature. This is quite ingenious and I can see the rationale for using a signature with a clearly defined pattern and one that has a rather random-looking pattern (SBS3 is indeed the best example for the latter). However, it was not clear to me why the authors chose to demonstrate this with SBS7a specifically, when other signatures are also well defined and further discussed in their manuscript (e.g. APOBEC, the MSI signature SBS44 etc.). It may be that the results obtained are very similar, but I would be curious to see a few more signatures in there (in fig 1c,e,f,g, or included separately in the supplementary material) so as to ensure it was not the case of simply showcasing the best results.

2. It is great to see that single base signatures are captured well following the application of the filters (fig 1 panels c-g). What about indel signatures? I can see that indel signatures are reported later on in the analyses, but I don't have any clear sense for how strong and reliable this signal is. I imagine there are very few indels that can be detected in each sample. The authors should provide evidence the indel signatures can also be captured reliably, at least to a certain extent, or otherwise discuss this as a limitation and propose ways to deal with this limitation in the future.

3. A key question is to what extent the signatures captured in ctDNA are faithful to the ones in the originating tumour if the tumour tissue were sequenced at high depth. The authors try to address this in two ways: (1) using one BRCA1 mutant breast cancer case where tumour and plasma have been sequenced at high depth, and compare to plasma-inferred signatures at low coverage; and (2) in low coverage plasma ctDNA in breast and colon cancer cohorts with annotation of BRCA and MSI status, respectively, and in melanoma where the UV signature would be expected. These are good ways to validate their inferred signatures, but (1) is only anecdotal in n=1 and (2) are imperfect proxies of the real signatures. Is there any larger cohort of plasma ctDNA and matched tumour tissue sequencing at higher depth that the authors could use to further validate their method? If not, the authors should acknowledge the aspects I pointed out as limitations of their method.

4. In figure 2f, there seem to be more BRCA-WT samples with SBS3 detected than BRCA-mutant samples (if considering absolute numbers, not percentages as in e). Similarly, in figure 2h, the SBS7 signatures are present only in a subset of samples, when most melanoma samples would likely present SBS7 to some extent. It is difficult to assess if the method is actually performing well without the availability of matched tumour tissue sequencing. The authors should comment on what they expect the sensitivity and specificity of their method might be, or ways to better address this in the future.

5. The authors should discuss how their tool compares against others available in the field for calling SNVs and indels from

low coverage ctDNA sequencing.

Reviewer #2

(Remarks to the Author)

The authors illustrate a computational pipeline for low coverage whole genome sequencing of circulating free DNA (cfDNA), which aims to detect circulating tumour DNA (ctDNA). The ctDNA detection mostly focuses on identifying somatic mutations that can be used to infer mutational signatures. Mutational signatures are patterns of somatic mutations that are related to specific mutational processes, which are linked to either environmental factors (e.g., UV damage) or DNA repair defects (e.g., mismatch repair deficiency or homologous recombination deficiency).

The performance of the proposed method, mostly sensitivity of detection of mutational signatures, is investigated, using both simulated and real data. The effect of various factors on the performance are also part of the investigation, with focus on mutation filtering, and minimum required coverage and purity for detection.

Overall, the article is well written and well structured. The strength of the presented method is its applicability to low coverage samples without the requirement of a paired normal sample. While the method is only applicable when the ctDNA is at a sufficiently high tumour purity, here estimated around 3%, there may be situations where high coverage WGS is not an option, and the described method can be extremely valuable.

While my assessment is overall favourable, I believe that the article in its current form is not yet ready for publication. There are a few points that I think require some attention, for example the effectiveness of the mutational signature analysis, and in some cases a possible lack of appropriate controls.

Major

1. The analysis of the BRCA1 mutant patient with sequencing from both solid tumour and plasma is limited. This analysis illustrates that a set of mutational signatures can be detected in both solid tumours using a traditional pipeline and in plasma using the proposed method. This result has a few limitations. A) it is an n=1; B) there is no control, such as a breast cancer with BRCA1 intact; C) SBS3, the most relevant signature detected here, has a relatively low specificity, as illustrated in figure 2f; D) it is lacking a comparative analysis of the actual somatic variants detected by the methods. E) it has a tumour purity of 60%, which is closer to a typical solid tumour sample than ctDNA in a plasma sample. For points A, B and E, I guess there is not much to do unless the authors have access to additional paired samples. For point C, my suggestion would be to look also at other signatures relevant to HR deficiency, such as SBS8 and deletion at MH, and provide a list of what likely false positive signatures were detected. For point D, I would be interested in seeing how many mutations are detected in the plasma sample using the proposed method and how many of these were in the solid tumour.

2. The analysis of melanoma samples in figure 2g,h seems to lack control samples and the UV damage related signature DBS1. I think it would be helpful to show here an appropriate control set to illustrate that UV damage related signatures are more likely to be present in melanoma than in other control samples (e.g. from a tissue not exposed to UV damage). I am surprised not to see DBS1 here among the enriched DBS signatures, which has been linked to UV damage. Notice that not all melanoma samples have UV damage, so showing that there are more samples with SBS7a/c and DBS1 in melanoma vs control should be enough.

3. In general, the mutational signature analysis seems to be tuned towards overfitting, leading to false positive calls of mutational signatures in the samples. This can be due to a few factors, such as the low number of somatic mutations detected in plasma and trying to fit many signatures instead of preselecting biologically relevant signatures (e.g. we don't expect the smoking signature SBS4 in breast cancer, yet according to the supplementary tables, most samples seem to have it). Please consider revising your signature analysis to improve specificity.

4. Perhaps something that is missing is how purity and coverage influence the number of detected somatic mutations, and in turn how many somatic mutations are then required for the detection of the mutational signatures. If possible please add at least some discussion of this.

Minor

5. In figure 1f, or in its legend, please provide the name of the 6 SBS signatures that are present in all 3 samples

6. Additional comparison/discussion of mutational signatures observed here with mutational signatures observed in published datasets could be useful to interpret the results. For example, something worth considering is the similarity between SBS7a and SBS2. They both have a strong component of C>T with T at the 5' of the mutation. According to published mutational signature datasets, SBS2 is found in a small proportion of melanoma samples, so the fact that SBS2 is detected in so many melanoma samples might indicate that the signature analysis could not properly differentiate between SBS2 and SBS7a, perhaps due to a low number of somatic mutations available. This is just a general suggestion to add more discussion if possible.

Reviewer #3

(Remarks to the Author)

The authors present a new single-read mutation detection method and software tool optimised for cell-free DNA, and use this approach to study mutational signatures in plasma samples. While previous studies have explored similar single-read mutation detection approaches (the authors cite two studies, but multiple other studies have explored this concept, e.g. [Zviran, Nature Med, 2020]), I applaud the authors for being the first to actually release/share the method as a software tool, well done! Their method is intuitive and straight-forward (essentially comprising a set of reasonable data/read filters), and the contribution from each filtering step is documented. The authors then explore if they can detect mutational signatures in cfDNA samples from cancer patients, a challenging task also explored in at least two previous studies. Their analysis are generally well designed and explained, and of higher quality than the previous studies in this domain, exploring both impact

of sequencing depth and ctDNA levels (Tumor Purity, TP). However, I still have some concerns relating to the use of controls and assumptions in their analysis (see below). Overall, it is important that the manuscript more clearly explores and thoroughly documents the limitations of the approach, so users don't blindly use this method on any lpWGS sample, tumor type, and mutational signature. It is a problem, that after reading the paper, I still don't fully understand how key parameters (1: sequencing coverage, 2: tumor purity, 3: mutational load of signature) influence the ability to detect a given signature in cfDNA. Furthermore, I have some concerns relating to the validation data presented for SBS2 and SBS3 in their paper.

Major issues

>> Assumptions for Fig 1C are not realistic

Fig. 1C explores the power to detect mutation signatures when spiking mutations of a given signature into a baseline sample. However, this analysis is based on an unrealistic high mutation load of the mutation signatures. For example, SBS3 is a signature enriched in some breast cancer tumors. When present, this signature contributes ~250 mutations in a breast cancer genome (~5% of TMB contributed by signature, ~5000 median mutations in genome, [Alexandrov et al., Nature, 2020]). However, the authors analysis is based on spiked-in variant counts ranging from 300 to 300,000, clearly outside the naturally occurring range. The authors should address this issue with a more detailed analysis, differentiating between signatures with a high and low mutational load. Moreover, the authors state that signal drops off at 5 muts/mb, does this mean that their approach cannot be used on tumors with mutation loads <5 muts/mb (most tumors)?

>> Control experiments restricted to a few mutational signatures (SBS7a and 3)

Why did the authors choose these two signatures for their preliminary analysis (Fig. 1)? The authors should replicate the analysis in Fig 1c-g for additional mutational signatures, for example just the first 20 signatures in COSMIC (to avoid selection bias).

>> Control experiment with matched breast tumor/plasma sample (Fig 1h,i)

The analysis of the breast cancer sample with matched tumor/plasma/germline sample could benefit from a deeper and more thorough analysis that could help readers understand the requirements and limitations of their approach. This sample constitutes an unusual combination of high sequencing coverage (9x) and very high tumor purity (60%!), making it a best-case scenario for plasma-based mutation signature detection. The authors should use their previously described approaches to dilute the TP (down to 3-5%) and coverage (down to 1x) of this sample, exploring when detection is not possible anymore and whether there are differences between the individual signatures (for example relating to mutation load of signature).

>> Additional validation of SBS2 signature

The finding that SBS2 weights can potentially stratify cancer from healthy samples in this cohort is intriguing. However, the authors should take steps to ensure that this is not related to a batch effect in the normal and cancer samples used in this single cohort. The authors should verify this discriminatory signal in additional cohorts of healthy and cancer lpWGS samples.

Furthermore, as important additional validation, could the authors verify that cancer types with known high SBS2 mutational load also have higher weights in their corresponding liquid biopsy samples? For example, how do the authors explain high SBS2 mutational signature weights in colorectal cancer liquid biopsy samples when SBS2 is absent in most colorectal cancer tumors according to the PCAWG group (Alexandrov et al., Nature, 2020, Fig 3)?

>> Additional validation of SBS3 signature

The finding that SBS3 can be found in BRCA-mutant breast cancer plasma samples is indeed interesting. However, I would again urge the authors to do further verification of this finding. One way to approach this could be to confirm that SBS3 weights are found elevated specifically in known SBS3-associated tumor subtypes. According to Alexandrov et al. 2020, almost all ovarian cancer tumors are strongly enriched for SBS3. The large lpWGS cohort from Fig. 2a-c could therefore be used to define positive (ovarian) and negative (select other cancer types) plasma sample sets for which we would expect to observe a strong difference in SBS3 weights.

>> High (9x) coverage used in figure 1c-g

This sample is at the limit of what one would classify as lpWGS/LCWGS in cfDNA (usually <3-5x). Are the results stable when downsampling to 3x?

>> Fig 1g poorly described in text

How was this done? The authors merely state "Using patient data, we next assessed MisMatchFinder's ...". Which and how many samples were used? How was dilution done? How do the authors know these signatures are present in the sample in the first place?

>> Concordance is unclear in Fig 1h

A quick look at the concordance between the stacked diagrams in Fig 1h suggest strong concordance between the tumor/normal and plasma-deepWGS/normal columns (A+B), with lower concordance for their method (column C). This aspect is not highlighted in the text. The authors should quantify the concordance in more detail. Additionally, why are only some signatures highlighted and used in this concordance analysis?

>> Potential for use of tumor/normal samples for additional controls

The authors could explore if they can use tumor samples to better characterise the limits of the mutational signature

detection. Tumor (and their matched normal/germline) samples have the advantage that the ground-truth mutational signatures can be established from the original sample based on accurate mutation calls (tumor vs. normal). The authors could then apply their cfDNA-tailored approach (perhaps without fragment length filter) to explore the detection power when inferring mutational signatures based on single reads in the tumor samples. These single-read results could then be compared to ground truth from standard tumor tissue mutation signature analysis. Furthermore, tumor samples could be downsampled (<10x) and diluted for lower tumor purity (using matched normal) to explore when mutational signatures cannot be reliably detected and quantified. In this testing framework, the authors could explore which types of signatures (e.g. high vs. low mutation load signature) can and cannot be detected with their approach.

Minor issues

>> “Unsupervised clustering between healthy and cancer samples using SBS signatures revealed APOBEC-linked SBS2 as a major discriminatory signature across most cancer types (Fig. 2b,c).”

This sentence appears to be inaccurate. Fig 2b is unsupervised clustering but does not show SBS signatures or how they discriminate between cancer and healthy samples. Fig 2c appears to be a supervised analysis (t-test) comparing SBS2 between cancer and healthy samples.

>> HRD and PARP inhibitors In the introduction the authors state the potential to use “PARP inhibitors in patients with HRD” signatures. However, it is not clear whether the authors were able to identify HRD signatures in the plasma samples?

>> Issues with external data cohort (EGAS00001003258)

The authors use an external cohort of 273 pan-cancer and 262 healthy LCWGS plasma samples. In the methods section it is stated that this data is obtained from the EGA project with accession EGAS00001003258. However, according to EGA page for this study (<https://ega-archive.org/studies/EGAS00001003258>), this project only contains 118 samples and 236 files.

Furthermore, the sample ID’s provided in Suppl. Table 3 (EGAFxxx) do not overlap with the sample IDs listed for this EGA project (<https://ega-archive.org/datasets/EGAD00001004939>). This is confusing, and I would encourage the authors to verify the provided data accession codes.

>> Data sharing

It is not clear if their internal cohorts used for validation (Fig 1h; Fig 2e,f) will be shared, to enable full transparency and reproducibility of their results.

Version 1:

Reviewer comments:

Reviewer #1

(Remarks to the Author)

The authors have addressed all my comments and I believe the manuscript is much improved. I have no further concerns and congratulate the authors for this nice work.

Reviewer #2

(Remarks to the Author)

I’m happy to report that the authors have done an excellent job addressing all my concerns. I consider all my comments resolved.

Reviewer #3

(Remarks to the Author)

The authors have addressed most of my comments. These two remaining aspects would further increase the quality and rigour of the manuscript, and make it easier for readers and users of the software to understand the assumptions and pitfalls of mutational signature analysis in ctDNA samples:

1) I appreciate the authors new Fig 1f+g exploring how coverage and tumor purity (ctDNA fraction) impacts the recovery of mutational signatures. However, this analysis doesn’t explore the (common) worst-case scenario where both of these factors are low (e.g. lpWGS coverage 1x and purity <5%). I would urge the authors to add analysis of more samples with less ideal combinations of coverage and purity in these figures. Please also list the mutational loads (mut/MB) for each of these signatures in their respective samples in the figure.

2) Under limitations and recommendations the authors state that: “. We recommend that at least 1x coverage for plasma sequencing data is essential for detecting mutational signatures with distinct profiles such as SBS2 and SBS7. However, for “flat”, non-distinctive signatures such as SBS3, higher sequencing depth (>3x) is necessary to ensure robust results,

particularly in samples with <10% TP.”

I recommend that this section also mention that it is more challenging to recover mutational signatures with low mutational burden (which their results also indicate), and that recovery is impacted by both sequencing coverage, sample purity (ctDNA levels), and tumor type (dictating expected signature load).

Accordingly, the 1x coverage recommendation should ideally be coupled with a recommendation for TP (x%) and minimum signature mutation load (XX mut/mb). “We recommend that at least 1x coverage and TP of X% for plasma sequencing of ... such as SBS2 and SBS7 with high mutational loads (XX mut/MB)” ...

Version 2:

Reviewer comments:

Reviewer #3

(Remarks to the Author)

The authors have addressed my last concerns, I appreciate the extra work and rigour.

RESPONSE TO REVIEWER COMMENTS

We thank all three reviewers for their thoughtful comments and constructive critique regarding our manuscript (NCOMMS-24-05248). We have fully addressed all the comments, concerns and suggestions, as described below in our detailed point-by-point description of the revised manuscript and believe that these updates have significantly strengthened our original manuscript.

Reviewer #1 (Remarks to the Author): Expert in computational cancer genomics, bioinformatics, and mutational signatures

Hollizeck et al present a compelling study demonstrating that mutational signatures can be inferred successfully from plasma circulating tumour DNA. They introduce a new tool MisMatchFinder that can call SNVs and indels from low coverage ctDNA sequencing and subsequently use established methods in the field (implemented in sigminer) to infer the mutational signatures in a variety of plasma ctDNA cohorts. The authors have paid particular attention to filtering mutation calls so as to ensure the optimal capture of cancer-related signal, and demonstrate this convincingly in the first part of the manuscript. The statistics are sound. I believe this is an important study opening up exciting possibilities in exploring mutational signatures for the early detection of cancer in liquid biopsies, and the authors provide some initial evidence this could be useful particularly for MSI high CRCs and HR deficient breast cancers. I have a few questions/comments that I believe would strengthen the study:

We thank the reviewer for a very constructive review of our manuscript and we are encouraged by the enthusiasm shown for our work.

1. The authors tested the effectiveness of their filters by spiking somatic variants linked to the SBS7a UV damage signature and SBS3 HRD signature. This is quite ingenious and I can see the rationale for using a signature with a clearly defined pattern and one that has a rather random-looking pattern (SBS3 is indeed the best example for the latter). However, it was not clear to me why the authors chose to demonstrate this with SBS7a specifically, when other signatures are also well defined and further discussed in their manuscript (e.g. APOBEC, the MSI signature SBS44 etc.). It may be that the results obtained are very similar, but I would be curious to see a few more signatures in there (in fig 1c,e,f,g, or included separately in the supplementary material) so as to ensure it was not the case of simply showcasing the best results.

We agree with the reviewer that it is important to show the generalisability of our method to other signatures. Therefore, in the revised manuscript, through *in-silico* analyses, we test MisMatchFinder on five additional signatures (SBS2, SBS4, SBS7c, SBS13, and SBS44) which all show consistent results to the original signatures SBS7a and SBS3 (Figure 1c and Supplementary Fig. 1). We also more comprehensively test detection of the SBS2 signature which is related to APOBEC enzymatic activity by analysing new matched tumour/germline and plasma data from a bladder cancer patient (Figure 1d,e,f,g and h). Furthermore, we have carried out these additional analyses on sequencing data with varying depths, from 9x to 0.1x, and across a range of tumour purities (10-1%) (Figures 1f and 1g, Supplementary Figs 3 and 4). We believe that these comprehensive additions provide further confidence in the accuracy, generalisability and robustness of our results.

2. It is great to see that single base signatures are captured well following the application of the filters (fig 1 panels c-g). What about indel signatures? I can see that indel signatures are

reported later on in the analyses, but I don't have any clear sense for how strong and reliable this signal is. I imagine there are very few indels that can be detected in each sample. The authors should provide evidence the indel signatures can also be captured reliably, at least to a certain extent, or otherwise discuss this as a limitation and propose ways to deal with this limitation in the future.

We agree with the reviewer that it would be ideal to validate the performance of MisMatchFinder on indels (ID) to a similar extent as we have showcased for SBS signatures by validating each filtering step using synthetic data. However, due to the complexity of the human genome, it is much more difficult to simulate indels for mutational signature analysis and be confident of the results^{1,2}. Indels do not emerge at random positions across the genome. Certain types of indels are enriched in regions with certain sequence context features, such as short tandem repeats and microhomologies. *In-silico* simulation of different ID signatures with these sequence context features could therefore easily be biased by how the indels are spiked in. We have therefore not undertaken *in-silico* simulation of different ID signatures and have discussed this as a limitation in the revised manuscript (page 4, lines 148-154).

However, we would like to reassure the reviewer that throughout the manuscript, when we report a signature as “detected” such as the indel signatures in Figure 2i, we have employed a large panel of healthy plasma controls (n=60) to obtain a limit of detection per signature. We only report signatures in cancer plasma datasets if they are above the 99% quantile of a beta distribution fitted to signature weights in the healthy cohort, to ensure stringent reporting for all SBS, DBS and ID signatures. We apologise if this information was unclear in the original manuscript. We have now further highlighted the use of healthy controls on page 2 lines 80-84, page 6 lines 283-291 and in Figure 1e. The technical details are presented in the Methods section.

3. A key question is to what extent the signatures captured in ctDNA are faithful to the ones in the originating tumour if the tumour tissue were sequenced at high depth. The authors try to address this in two ways: (1) using one BRCA1 mutant breast cancer case where tumour and plasma have been sequenced at high depth, and compare to plasma-inferred signatures at low coverage; and (2) in low coverage plasma ctDNA in breast and colon cancer cohorts with annotation of BRCA and MSI status, respectively, and in melanoma where the UV signature would be expected. These are good ways to validate their inferred signatures, but (1) is only anecdotal in n=1 and (2) are imperfect proxies of the real signatures. Is there any larger cohort of plasma ctDNA and matched tumour tissue sequencing at higher depth that the authors could use to further validate their method? If not, the authors should acknowledge the aspects I pointed out as limitations of their method.

We thank the reviewer for acknowledging the different approaches we have used to assess if low-coverage ctDNA sequencing closely captures the true signal from tumour tissue with respect to mutational signatures. We agree with the reviewer that the results we previously presented looking at direct signature concordance between matched tumour/germline samples and plasma data were limited as they were confined to a single patient.

In the revised manuscript, we have now included an additional patient with bladder cancer where sequencing of temporally matched fresh tumour tissue, germline DNA and plasma was possible, revealing the presence of several mutational signatures related to APOBEC activity.

This bladder cancer patient along with our previous breast cancer patient with a confirmed *BRCA* mutation provide us with high coverage, tumor/germline and plasma sequencing data to validate our results in both a “flat” (SBS3) versus “peaked” (SBS2) signature context. We have now used these samples to comprehensively test the fragmentomics filter of

MisMatchFinder (Figure 1d), and to study the effect of tumour purity (Figure 1f) and sequencing depth (Figure 1g) on the limits of detection of our method.

To comprehensively assess the concordance between signatures captured in ctDNA compared to those detected in the tumour, we first called somatic variants from high depth, tumour/germline sequencing data of each patient with the DRAGEN pipeline (v4.2.4, https://support-docs.illumina.com/SW/dragen_v42/Content/SW/DRAGEN/GPipelineIntro_fDG.htm)³. The output VCF files from DRAGEN were further processed through an in-house variant filtering and curation workflow to prioritize small somatic variant calls: (<https://github.com/umccr/umccrise/tree/master>). Then both SNV and indel results were fitted to mutational signatures (COSMIC v3.2) with sigminer (v2.3.0)⁴. The same pipeline was used to call somatic variants and signatures from the high-coverage plasma/germline datasets. The signatures of each sample were then intersected with signatures previously found in bladder and breast cancers respectively (Alexandrov et al. 2020⁵) to assess tumour-type specific concordance. We then compared these results to the signatures obtained using our MisMatchFinder analysis on plasma low coverage WGS data without use of the germline control (Fig. 1h and 1i).

These new results show that the MisMatchFinder approach effectively captures the main signatures in tumour tissue with high cosine similarities (>0.662-0.999) to quantify this concordance. We have performed the tumour/plasma concordance analysis across varying sequencing depths (9x down to 0.1x) and tumour purities (10% down to 1%) to show that the key mutational signatures remain detectable and consistent even at extreme values to provide further confidence in our results (Supplementary Fig 4).

4. In figure 2f, there seem to be more BRCA-WT samples with SBS3 detected than BRCA-mutant samples (if considering absolute numbers, not percentages as in e). Similarly, in figure 2h, the SBS7 signatures are present only in a subset of samples, when most melanoma samples would likely present SBS7 to some extent. It is difficult to assess if the method is actually performing well without the availability of matched tumour tissue sequencing. The authors should comment on what they expect the sensitivity and specificity of their method might be, or ways to better address this in the future.

The reviewer has raised an important point that SBS3 is being detected in patients without BRCA mutations and that it is difficult to assess if these were true or false positives without matched tumour tissue sequencing.

In the revised manuscript, there have now been major updates to this analysis. We have now employed updated versions of sigminer (v.2.3.0) and COSMIC (v3.2). All signature weights have also been normalised to remove possible sequencing artefact related signatures. Throughout the manuscript when we report a signature as “detected”; we have employed a large panel of healthy plasma controls (n=60) to obtain a limit of detection per signature and we have only reported signatures in any plasma dataset if they are above the background observed in the healthy controls. In the original manuscript, our detection limit for each signature was set at the 95% quantile of the beta distribution fitted to the signature weights in our healthy control panel. In the revised manuscript, to ensure even greater stringency in our analysis, we have now only reported signatures as “detected” if they are above the 99% quantile observed across the healthy cohort. These revisions have been discussed in detail on page 2 lines 80-84, page 6 lines 283-291, Figure 1e. and in the Methods section.

To further assess the sensitivity and specificity of our method to detect the SBS3 signature as a surrogate for HRD, we have now compared it directly with the performance of shallowHRD⁶ (v1.13), a HRD detecting tool specifically designed for low coverage sequencing data. While

this tool was developed for tumour sequencing and relies on detection of large scale genomic alterations characteristic of HRD, it is the only HRD detection tool tailored for low coverage sequencing data which could be used as orthogonal validation to MisMatchFinder in this context. For the six breast cancer samples with known *BRCA1/2* mutant status, which were expected to demonstrate a HRD profile, our tool detected the SBS3 signature across all samples (100% sensitivity) while shallowHRD only detected five out of the six cases (83% sensitivity). MisMatchFinder also detected an SBS3 signature in 8/39 patients without a germline *BRCA* mutation, and this was concordant with the detection of HRD via shallow HRD in 6/8 of these cases (Supplementary Fig. 5), noting that breast cancer patients without germline *BRCA* mutations may have homologous recombination deficiency originating by other mechanisms⁷. In parallel, we have now also looked at deletions within microhomology regions for the 6 *BRCA* mutation cases and the 39 *BRCA* wild-type cases. As expected, we observed that the cases with a known *BRCA* germline mutation had the highest number of microhomology deletions (Supplementary Fig. 6). Importantly, the 8 *BRCA* wild-type cases which showed an elevated SBS3 mutational signature profile using MisMatchFinder, also had an elevated number of microhomology deletions compared to *BRCA* wild type cases without detection of the SBS3 signature (Mann–Whitney U test, P-value=0.023), adding further evidence for the presence of HRD in these cases (Supplementary Fig.6).

The reviewer has also commented on the fact that SBS7 was present in only a subset of samples in Figure 2h. In our revised analysis with our stringent detection threshold (above the 99% quantile of a beta distribution per signature observed across the healthy controls), the UV-related signatures (SBS7a, SBS7b, and DBS1) were observed in 28% cases (11/40) across our melanoma cohort. Of the 29 cases which did not show evidence of a UV-related signature, 17/29 (59%) had no detectable ctDNA according to ichorCNA (<3% TP). Moreover, previous studies have also shown that not all cutaneous melanomas harbour high SBS7 signatures, and this may be a contributing factor to the decrease in UV signature detection in low TP samples⁸.

5. The authors should discuss how their tool compares against others available in the field for calling SNVs and indels from low coverage ctDNA sequencing.

We thank the reviewer for this suggestion. Based on our current knowledge, there are currently no computational tools that are specifically designed to call SNVs and indels from low coverage sequencing of circulating tumour DNA. However, we believe that (i) the further benchmarking we have performed across additional samples and (ii) our comparisons to alternative tools such as shallowHRD, strengthen the confidence in MisMatchFinder's accuracy.

Reviewer #1 (Remarks on code availability): While I have not run the code myself, it seems to be well documented and easy to run, with example files provided. The programming language used is not one of the most commonly used ones, but the authors have put extra effort in simplifying the process for the user, such that it seems a single line of code is sufficient to run the entire tool. Overall, very user-friendly.

We appreciate the reviewer's positive assessment of our code base. We are eager to share our tools with the research community and the ease-of-implementation of these tools is very important to us.

Reviewer #2 (Remarks to the Author): Expert in computational cancer genomics, bioinformatics, and mutational signatures

The authors illustrate a computational pipeline for low coverage whole genome sequencing of circulating free DNA (cfDNA), which aims to detect circulating tumour DNA (ctDNA). The

ctDNA detection mostly focuses on identifying somatic mutations that can be used to infer mutational signatures. Mutational signatures are patterns of somatic mutations that are related to specific mutational processes, which are linked to either environmental factors (e.g., UV damage) or DNA repair defects (e.g., mismatch repair deficiency or homologous recombination deficiency).

The performance of the proposed method, mostly sensitivity of detection of mutational signatures, is investigated, using both simulated and real data. The effect of various factors on the performance are also part of the investigation, with focus on mutation filtering, and minimum required coverage and purity for detection.

Overall, the article is well written and well structured. The strength of the presented method is its applicability to low coverage samples without the requirement of a paired normal sample. While the method is only applicable when the ctDNA is at a sufficiently high tumour purity, here estimated around 3%, there may be situations where high coverage WGS is not an option, and the described method can be extremely valuable.

While my assessment is overall favourable, I believe that the article in its current form is not yet ready for publication. There are a few points that I think require some attention, for example the effectiveness of the mutational signature analysis, and in some cases a possible lack of appropriate controls.

We appreciate the reviewer's thoughtful feedback and welcome their comment on how valuable the approach described in our study could be.

Major

1. The analysis of the BRCA1 mutant patient with sequencing from both solid tumour and plasma is limited. This analysis illustrates that a set of mutational signatures can be detected in both solid tumours using a traditional pipeline and in plasma using the proposed method. This result has a few limitations. A) it is an n=1; B) there is no control, such as a breast cancer with BRCA1 intact; C) SBS3, the most relevant signature detected here, has a relatively low specificity, as illustrated in figure 2f; D) it is lacking a comparative analysis of the actual somatic variants detected by the methods. E) it has a tumour purity of 60%, which is closer to a typical solid tumour sample than ctDNA in a plasma sample. For points A, B and E, I guess there is not much to do unless the authors have access to additional paired samples. For point C, my suggestion would be to look also at other signatures relevant to HR deficiency, such as SBS8 and deletion at MH, and provide a list of what likely false positive signatures were detected. For point D, I would be interested in seeing how many mutations are detected in the plasma sample using the proposed method and how many of these were in the solid tumour.

The reviewer has brought up several important issues and we have addressed all of them as described below:

(i) Reviewer comments A-E

In the revised manuscript, we have now included an additional patient with bladder cancer where sequencing of temporally matched fresh tumour tissue, germline DNA and plasma (tumour purity 12%) was possible, revealing the presence of several mutational signatures related to APOBEC activity. This bladder cancer patient along with our previous breast cancer patient with a confirmed *BRCA* mutation provide us with high coverage, tumour/germline and plasma sequencing data to validate our results in both a "flat" (SBS3) versus "peaked" (APOBEC) signature context. We have now used these samples to comprehensively test the fragmentomics filter of MisMatchFinder (Figure 1d), and to study the effect of tumour purity (Figure 1f) and sequencing depth (Figure 1g) on the limits of detection of our method.

To comprehensively assess the concordance between signatures captured in ctDNA compared to those detected in the tumour, we first called somatic variants from high-coverage, tumour/germline sequencing data of each patient with the DRAGEN pipeline (v4,2,4, https://support-docs.illumina.com/SW/dragen_v42/Content/SW/DRAGEN/GPipelineIntro_fDG.htm)³. The output VCF files from DRAGEN were further processed through an in-house post processing workflow to prioritize small somatic variant calls (<https://github.com/umccr/umccrise/tree/master>).

Then both SNV and indel results were fitted to mutational signatures (COSMIC v3.2) with sigminer (v2.3.0)⁴. The same pipeline was used to call somatic variants and signatures from the high-coverage plasma/germline datasets. The signatures of each sample were then intersected with signatures previously found in bladder and breast cancers in large-scale patient cohorts (Alexandrov et al. 2020⁵) to assess tumour-type specific concordance. We then compared these results to the signatures obtained using our MisMatchFinder analysis on plasma low coverage WGS data without use of the germline control (Fig. 1h and 1i). These new results show that the MisMatchFinder approach effectively captures the main signatures in tumour tissue with high cosine similarities (>0.662-0.999) to quantify this concordance.

As suggested by the reviewer, we have also performed the tumour/plasma concordance analysis across varying sequencing depths (9x down to 0.1x) and tumour purities (10% down to 1%) to show that the key mutational signatures remain detectable and consistent even at extreme values to provide further confidence in our results (Supplementary Fig 4).

(ii) Reviewer comment C

In the revised manuscript, we have undertaken several additional analyses to assess the specificity of our SBS3 signature detection. In the revised manuscript, there have now been major updates to this analysis. We have now employed updated versions of sigminer (v.2.3.0) and COSMIC (v3.2). All signature weights have also been re-normalised after removing possible sequencing artefact related signatures.

Throughout the manuscript when we report a signature as “detected”; we have employed a large panel of healthy plasma controls (n=60) to obtain a limit of detection per signature and we have only reported signatures in any plasma dataset if they are above the background level observed in the healthy controls. In the original manuscript, we fitted weights of each signature to a beta distribution and our detection limit for each signature was set at the 95% quantile of the beta distribution derived from our healthy control panel. In the revised manuscript, to ensure even greater stringency in our analysis, we have now only reported signatures as “detected” if they are above the 99% quantile of the beta distribution observed across the healthy cohort. This revision has been discussed in detail on page 2 lines 80-84, page 6 lines 283-291, Figure 1e. and in the Methods section.

To further assess the sensitivity and specificity of our method to detect the SBS3 signature as a surrogate for HRD, we compared it directly with the performance of shallowHRD⁶ (v1.13), a HRD detecting tool specifically designed for low coverage sequencing data. While this tool was developed for tumour tissue sequencing and relies on detection of large scale genomic alterations characteristic of HRD, it is the only HRD detecting tool tailored for low coverage sequencing data which could be used as orthogonal validation to MisMatchFinder in this context. For the six breast cancer samples with known *BRCA1/2* mutant status (*BRCA*-mutant), which were expected to demonstrate a HRD profile, our tool detected the SBS3 signature across all samples (100% sensitivity) while shallowHRD only detected five out of the six cases (83% sensitivity). MisMatchFinder also detected an SBS3 signature in 8/39 patients without a germline *BRCA* mutation, and this was concordant with the detection of HRD via shallow HRD in 6/8 of these cases (Supplementary Fig. 5), noting that breast cancer patients

without germline *BRCA* mutations may have homologous recombination deficiency originating by other mechanisms⁷.

Finally, as suggested by the reviewer, we have also looked at deletions with microhomology (MH) within our six *BRCA*-mutant and 39 *BRCA*-WT samples. We used VarSCAT² to label deletions within microhomology based on the definition from HRDetect^{9,10}. As expected, we observed that the *BRCA*-mutant samples had the highest number of MH deletions. Importantly, the 8 *BRCA*-WT samples which showed an elevated SBS3 mutational signature profile using MisMatchFinder, also had an elevated number of MH deletions compared to *BRCA*-WT samples with no SBS3 signature detected, adding further evidence for the presence of HRD in these cases (Supplementary Fig.6, Mann-Whitney U test, p-value=0.02272).

(iii) Reviewer comment D

MisMatchFinder detects a greater number of mutations than the corresponding somatic variants called through standard tumour/germline mutation calling. In our analyses, standard tumour/germline somatic variant calling was performed using the DRAGEN pipeline and highly curated to avoid false positive results. In contrast, MisMatchFinder identifies a high number of “mismatches” as analysis is performed in a read-centric level, and it does not utilise a matched germline control. However, in our matched tumour/plasma datasets (for both our breast and bladder cancer patients), we further show that even though the number of detected variants by MisMatchFinder and those from tumour/germline somatic variant calling are different, the concordance for signature detection (based on cosine similarities of signatures) remains high (Fig 1h and 1i). We have performed the tumour/plasma concordance analysis across varying sequencing depths (9x down to 0.1x) and tumour purities (10% down to 1%) to show that the key mutational signatures remain detectable and consistent even at extreme values to provide further confidence in our results (Supplementary Fig 4).

2. The analysis of melanoma samples in figure 2g,h seems to lack control samples and the UV damage related signature DBS1. I think it would be helpful to show here an appropriate control set to illustrate that UV damage related signatures are more likely to be present in melanoma than in other control samples (e.g. from a tissue not exposed to UV damage). I am surprised not to see DBS1 here among the enriched DBS signatures, which has been linked to UV damage. Notice that not all melanoma samples have UV damage, so showing that there are more samples with SBS7a/c and DBS1 in melanoma vs control should be enough.

We apologise for the lack of clarity in how we use control samples. Throughout the manuscript when we report a signature as “detected”; we have employed a large panel of healthy plasma controls (n=60) to obtain a limit of detection per signature and we have only reported signatures if they are above the background observed in the healthy controls. Specifically in Figures 2e-j, whether there are two groups of cancer patients being compared (i.e with and without *BRCA* mutations) or a single cohort (patients with melanoma), we only report signatures if they are above the background observed in the healthy controls. In the original manuscript, our detection limit was set above the 95% quantile of a beta distribution per signature derived from our healthy control panel. However, in the revised manuscript, to ensure even greater stringency, we have now only reported signatures as “detected” if they are above the 99% quantile of the beta distribution fitted to the healthy cohort. This revision has been discussed in detail on page 2 lines 80-84, page 6 lines 283-291, Figure 1e and in the Methods section.

We agree with the reviewer that DBS1 is an important UV-damage signature. In the revised manuscript, with our more stringent detection threshold, we now report the top 4 signatures and show that DBS1 is the 2nd most abundant signature in our melanoma cohort (Fig. 2g).

3. In general, the mutational signature analysis seems to be tuned towards overfitting, leading to false positive calls of mutational signatures in the samples. This can be due to a few factors, such as the low number of somatic mutations detected in plasma and trying to fit many signatures instead of preselecting biologically relevant signatures (e.g. we don't expect the smoking signature SBS4 in breast cancer, yet according to the supplementary tables, most samples seem to have it). Please consider revising your signature analysis to improve specificity.

We believe this comment arises from how we have presented data. In the previous supplementary tables, we reported weights for every signature in every sample (except for sequencing error related signatures), even if these signatures were not detected above the background threshold set using the healthy controls. We have now updated this in the revised manuscript and we now only report a signature as “detected” if it is above the 99% quantile of a beta distribution per signature observed across the healthy control cohort. This shows that whilst the smoking signature SBS4 has “non-zero” weights in breast cancer, this signature is considered “not detected” in these samples because it is not above the background signal observed from the healthy controls. Supplementary tables 1-2 have now been updated to indicate “detected” and “not detected” signatures for each cancer sample.

4. Perhaps something that is missing is how purity and coverage influence the number of detected somatic mutations, and in turn how many somatic mutations are then required for the detection of the mutational signatures. If possible please add at least some discussion of this.

We thank the reviewer for this comment and hope that the additional results provided (Supplementary Fig 4) for the tumour/plasma concordance analysis across varying sequencing depths (9x down to 0.1x) and tumour purities (10% down to 1%) addresses this concern. In addition, we have expanded our *in silico* analysis for different mutational burdens across several signatures to assess the impact of the number of mutations on signature detection (Supplementary Fig 1). Finally, we have added further discussion regarding the recommended sequencing coverage required to detect different signatures at different tumour purity contexts, using our method (page 4, lines 164-167).

Minor

5. In figure 1f, or in its legend, please provide the name of the 6 SBS signatures that are present in all 3 samples

We apologize that our previous figure was not clear. In the revised manuscript, we have refined the figure and showed the signature concordance between tumour and plasma of the bladder cancer and breast cancer case. We now provide the exact signatures investigated in the figure legend of Figure 1h and 1i.

6. Additional comparison/discussion of mutational signatures observed here with mutational signatures observed in published datasets could be useful to interpret the results. For example, something worth considering is the similarity between SBS7a and SBS2. They both have a strong component of C>T with T at the 5' of the mutation. According to published mutational signature datasets, SBS2 is found in a small proportion of melanoma samples, so the fact that SBS2 is detected in so many melanoma samples might indicate that the signature analysis could not properly differentiate between SBS2 and SBS7a, perhaps due to a low number of somatic mutations available. This is just a general suggestion to add more discussion if possible.

We appreciate the reviewer for bringing out this interesting point. However, it should be noted that MisMatchFinder does not detect a low number of somatic mutations. In contrast, MisMatchFinder detects a greater number of “mismatches” than the corresponding somatic variants identified through standard tumour/germline mutation calling because the analysis is based on low coverage WGS data, and it does not utilise a matched germline control.

In the revised manuscript, we have also undertaken several additional steps to improve the specificity of signature detection. We have now employed updated versions of sigminer (v.2.3.0) and COSMIC (v3.2) and all signature weights have also been re-normalised after removing possible sequencing artefact related signatures. Whilst it is possible that our method may have difficulty differentiating between SBS2 and SBS7a, our detailed analysis of the bladder cancer patient with paired tumour and plasma data (where SBS2 is clearly detected at low purity/coverage combinations) and our melanoma cohort (where SBS7a is the top signature detected) suggests that MisMatchFinder can discriminate effectively between SBS2 and SBS7a. We include further discussion on this point on page 4 lines 154-167 in the revised manuscript.

Reviewer #3 (Remarks to the Author): Expert in computational cancer genomics, bioinformatics, variant calling methods, and ctDNA

The authors present a new single-read mutation detection method and software tool optimised for cell-free DNA, and use this approach to study mutational signatures in plasma samples. While previous studies have explored similar single-read mutation detection approaches (the authors cite two studies, but multiple other studies have explored this concept, e.g. [Zviran, Nature Med, 2020]), I applaud the authors for being the first to actually release/share the method as a software tool, well done! Their method is intuitive and straight-forward (essentially comprising a set of reasonable data/read filters), and the contribution from each filtering step is documented. The authors then explore if they can detect mutational signatures in cfDNA samples from cancer patients, a challenging task also explored in at least two previous studies. Their analysis are generally well designed and explained, and of higher quality than the previous studies in this domain, exploring both impact of sequencing depth and ctDNA levels (Tumor Purity, TP). However, I still have some concerns relating to the use of controls and assumptions in their analysis (see below). Overall, it is important that the manuscript more clearly explores and thoroughly documents the limitations of the approach, so users don't blindly use this method on any lpWGS sample, tumor type, and mutational signature. It is a problem, that after reading the paper, I still don't fully understand how key parameters (1: sequencing coverage, 2: tumor purity, 3: mutational load of signature) influence the ability to detect a given signature in cfDNA. Furthermore, I have some concerns relating to the validation data presented for SBS2 and SBS3 in their paper.

We thank the reviewer for their comprehensive review. We are heartened by their positive commentary on our clear experimental design, the specific testing of different analysis steps, the provision of software that can be easily implemented by others and the overall high quality of the study. In the revised manuscript, we have performed additional analyses to understand the effect of sequencing coverage, tumour purity and mutational load and these results are detailed below. Finally, we have also provided additional discussion on how these factors influence signature detection and clear recommendations on limits of detection in the revised manuscript (Pages 2-4).

Major issues

>> Assumptions for Fig 1C are not realistic

Fig. 1C explores the power to detect mutation signatures when spiking mutations of a given

signature into a baseline sample. However, this analysis is based on an unrealistic high mutation load of the mutation signatures. For example, SBS3 is a signature enriched in some breast cancer tumors. When present, this signature contributes ~250 mutations in a breast cancer genome (~5% of TMB contributed by signature, ~5000 median mutations in genome, [Alexandrov et al., Nature, 2020]). However, the authors analysis is based on spiked-in variant counts ranging from 300 to 300,000, clearly outside the naturally occurring range. The authors should address this issue with a more detailed analysis, differentiating between signatures with a high and low mutational load. Moreover, the authors state that signal drops off at 5 muts/mb, does this mean that their approach cannot be used on tumors with mutation loads <5 muts/mb (most tumors)?

This panel (Fig 1c) showcases analysis that determines the necessity of the germline filter. This is a very stringent but necessary filter to observe the signal across multiple mutational signatures. In our revised manuscript, we directly mimic the SBS3 mutational loads based on the COSMIC database (<https://cancer.sanger.ac.uk/signatures/sbs/sbs3/>) and investigate tumour mutational burdens at 0.05, 0.1, 0.25, 0.5, 0.75, 1, 5, 10 mutations / mb (Supplementary Fig. 1b). In parallel, we have assessed the detection rate of the SBS7a signature and five additional signatures (SBS2, SBS4, SBS7c, SBS13, and SBS44) across various tumour mutational burdens based on the COSMIC database (Supplementary Fig. 1). We show that we can detect these well-defined signatures down to 0.1 mutations per megabase at 3x sequencing coverage.

However, the above analysis, performed on synthetic data, cannot stand as a limit of detection assessment. We have more comprehensively investigated the sensitivity and specificity of MisMatchFinder using cancer patient samples in Fig 1f,g,h, and i where we have shown that MisMatchFinder can detect key mutational signatures across varying sequencing depths (9x down to 0.1x) and tumour purities (10% down to 1%).

To specifically assess the mutation burden required for detection of SBS3 with MisMatchFinder, we used our breast cancer cohort with known *BRCA1/2* mutations (n=6). The results showed that MisMatchFinder can detect SBS3 from all the *BRCA*-mutant samples with SBS3 mutation burdens ranging from 1.01-10.33 muts/mb (reviewer only table below).

Table 1. SBS3 signatures and meta data of *BRCA*-mutant samples.

Samples	BRCA status	SBS3 weights	SBS3 mutations	SBS3 muts/mb	Tumor purity
MBCB195	BRCA mutant	0.0765	28254	9.42	13%
MBCB268	BRCA mutant	0.1038	26392	8.80	42%
MBCB289	BRCA mutant	0.0459	3944	1.31	12%
MBCB314	BRCA mutant	0.0275	3020	1.01	11%
MBCB355	BRCA mutant	0.0767	30991	10.33	86%
MBCB401	BRCA mutant	0.0556	4656	1.55	31%

>> Control experiments restricted to a few mutational signatures (SBS7a and 3) Why did the authors choose these two signatures for their preliminary analysis (Fig. 1)? The authors should replicate the analysis in Fig1c-g for additional mutational signatures, for example just the first 20 signatures in COSMIC (to avoid selection bias).

We agree with the reviewer that it is important to show the generalisability of our method to other signatures. Therefore, in the revised manuscript, through *in-silico* analysis we test MisMatchFinder on five additional signatures (SBS2, SBS4, SBS7c, SBS13, and SBS44) which all show consistent results to the original signatures SBS7a and SBS3 (Figure 1c and Supplementary Fig. 1). We also more comprehensively test detection of the SBS2 signature which is related to APOBEC enzymatic activity by analysing new matched tumour/germline and plasma data from a bladder cancer patient (Figure 1d,e, f,g,and h). Furthermore, we have carried out these additional analyses on sequencing data with varying depths, from 9x to 0.1x, and across a range of tumour purities (10-1%) (Figures 1f and 1g, Supplementary Figs 3,4). We hope that these additions provide further confidence in the accuracy, generalisability and robustness of our results.

>> *Control experiment with matched breast tumor/plasma sample (Fig 1h,i)*
The analysis of the breast cancer sample with matched tumor/plasma/germline sample could benefit from a deeper and more thorough analysis that could help readers understand the requirements and limitations of their approach. This sample constitutes an unusual combination of high sequencing coverage (9x) and very high tumor purity (60%!), making it a best-case scenario for plasma-based mutation signature detection. The authors should use their previously described approaches to dilute the TP (down to 3-5%) and coverage (down to 1x) of this sample, exploring when detection is not possible anymore and whether there are differences between the individual signatures (for example relating to mutation load of signature).

In the revised manuscript, we have now included an additional patient with bladder cancer where sequencing of temporally matched fresh tumour tissue, germline DNA and ctDNA (tumour purity 12%) was possible, revealing the presence of several mutational signatures related to APOBEC activity. This bladder cancer patient along with our previous breast cancer patient with a confirmed *BRCA* mutation provide us with high coverage tumour/germline and plasma sequencing data to validate our results in both a “flat” (SBS3) versus “peaked” (APOBEC) signature context. As suggested by the reviewer, we have now used these samples to comprehensively test the fragmentomics filter of MisMatchFinder (Figure 1d), and to study the effect of tumour purity (Figure 1f) and sequencing depth (Figure 1g) on the limits of detection of our method.

To comprehensively assess the concordance between signatures captured in ctDNA compared to those detected in the tumor, we first called somatic variants from high depth, tumor/germline sequencing data of each patient with the DRAGEN pipeline ((v4.2.4, https://support-docs.illumina.com/SW/dragen_v42/Content/SW/DRAGEN/GPipelineIntro_fDG.htm)³. The output VCF files from DRAGEN were further processed through an in-house post processing workflow to prioritize small somatic variant calls.

All details can be found with the code: <https://github.com/umccr/umccrise/tree/master>

Then both SNV and indel results were fitted to mutational signatures (COSMIC v3.2) with sigminer (v2.3.0)⁴. The same pipeline was used to call somatic variants and signatures from the high-coverage plasma/germline datasets. The signatures of each sample were then intersected with signatures previously found in bladder and breast cancers respectively (Alexandrov et al. 2020⁵) to assess tumour-type specific concordance. We then compared these results to the signatures obtained using our MisMatchFinder analysis on plasma low coverage WGS data without use of the germline control (Fig. 1h and 1i).

These new results show that the MisMatchFinder approach effectively captures the main signatures in tumour tissue with high cosine similarities (>0.662-0.999) to quantify this concordance. We have performed the tumour/plasma concordance analysis across varying

sequencing depths (9x down to 0.1x) and tumour purities (10% down to 1%) to show that the key mutational signatures remain detectable and consistent even at extreme values to provide further confidence in our results (Supplementary Fig 4).

>> *Additional validation of SBS2 signature*

The finding that SBS2 weights can potentially stratify cancer from healthy samples in this cohort is intriguing. However, the authors should take steps to ensure that this is not related to a batch effect in the normal and cancer samples used in this single cohort. The authors should verify this discriminatory signal in additional cohorts of healthy and cancer lpWGS samples. Furthermore, as important additional validation, could the authors verify that cancer types with known high SBS2 mutational load also have higher weights in their corresponding liquid biopsy samples? For example, how do the authors explain high SBS2 mutational signature weights in colorectal cancer liquid biopsy samples when SBS2 is absent in most colorectal cancer tumors according to the PCAWG group (Alexandrov et al., Nature, 2020, Fig 3)?

We thank the reviewer for pointing out this issue. We acknowledge that elevated SBS2 weights were identified in colorectal cancer samples from the DELFI cohort (Fig 2c). However, in addition to the work of Alexandrov et al., Nature, 2020⁵, there are other studies which have analysed the mutational signatures of different cancer types. In particular, the Signal database (Degasperi et al., Nature cancer, 2020¹¹, <https://signal.mutationalsignatures.com/explore/cancer>), which contained mutational signature data across various cancer types, showed that 16% (383/2348) of colorectal cancer samples contained the SBS2 signature.

In addition, it is important to note that our SBS2 results are not confined to the DELFI cohort. Our revised manuscript has now evaluated SBS2 detection in data from bladder cancer and breast cancer patients with matched tissue and plasma (Fig 1h-i) as well as additional breast, melanoma and colorectal cancer cohorts (Fig 2e-j). In our revised analysis we have employed updated versions of sigminer (v.2.3.0) and COSMIC (v3.2). All signature weights have also been normalised to remove possible sequencing artefact related signatures. Finally, we now only report a signature as “detected” if it is above the 99% quantile of a beta distribution per signature observed across the healthy control cohort (n=60). Through this revised analysis, we have shown that SBS2 remains a dominant signature in the bladder cancer (Fig 1h) and breast cancer cases (Fig 1i), but not in the melanoma and colorectal cancer cohorts (Fig 2g-j) as expected according to the literature (Alexandrov et al Nature 2020⁵).

Whilst our results show clear identification of the SBS2 signature across multiple samples, we agree with the reviewer that additional validation of mutational signatures in liquid biopsy samples should be done in the future, and we further discuss this in the revised manuscript (page 4, lines 157-160).

>> *Additional validation of SBS3 signature*

The finding that SBS3 can be found in BRCA-mutant breast cancer plasma samples is indeed interesting. However, I would again urge the authors to do further verification of this finding. One way to approach this could be to confirm that SBS3 weights are found elevated specifically in known SBS3-associated tumor subtypes. According to Alexandrov et al. 2020, almost all ovarian cancer tumors are strongly enriched for SBS3. The large lpWGS cohort from Fig. 2a-c could therefore be used to define positive (ovarian) and negative (select other cancer types) plasma sample sets for which we would expect to observe a strong difference in SBS3 weights.

We agree with the reviewer that further validation of the SBS3 signature is important. However, the DELFI cohort from Fig. 2a-c is not ideal for this purpose as the data is only 1-2x coverage, making reliable identification of all SBS3 cases challenging, as highlighted above.

We have therefore taken a different approach. To further assess the sensitivity and specificity of our method to detect the SBS3 signature as a surrogate for HRD, we compared it directly with the performance of shallowHRD⁶ (v1.13), a HRD detecting tool specifically designed for low coverage sequencing data. While this tool was developed for tumour sequencing and relies on detection of large-scale genomic alterations characteristic of HRD, it is the only HRD detecting tool tailored for low coverage sequencing data which could be used as orthogonal validation to MisMatchFinder in this context. For the six breast cancer samples with known *BRCA1/2* mutant status, which were expected to demonstrate a HRD profile, our tool detected the SBS3 signature across all samples (100% sensitivity) while shallowHRD only detected five out of the six cases (83% sensitivity).

MisMatchFinder also detected an SBS3 signature in 8/39 patients without a germline *BRCA* mutation, and this was concordant with the detection of HRD via shallow HRD in 6/8 of these cases (Supplementary Fig. 5), noting that breast cancer patients without germline *BRCA* mutations may have homologous recombination deficiency originating by other mechanisms⁷.

In parallel, we have now also looked at deletions within microhomology regions for the 6 *BRCA* mutation cases and the 39 *BRCA* wild-type cases. As expected, we observed that the cases with a known *BRCA* germline mutation had the highest number of microhomology deletions (Supplementary Fig. 6). Importantly, the 8 *BRCA* wild-type cases which showed an elevated SBS3 mutational signature profile using MisMatchFinder, also had an elevated number of microhomology deletions compared to *BRCA* wild type cases with no SBS3 detected (Mann–Whitney U test, P-value=0.023), adding further evidence for the presence of HRD in these cases (Supplementary Fig.6).

>> *High (9x) coverage used in figure 1c-g*

This sample is at the limit of what one would classify as lpWGS/LCWGS in cfDNA (usually <3-5x). Are the results stable when downsampling to 3x?

We agree with the reviewer regarding the high coverage originally used in Figure 1c-g. In our revised manuscript, the sequencing coverage of data used for Figure 1c, 1d, 1f, 1h, and 1i is now 3x which is considered more representative of LCWGS. We have also tested a range of sequencing depths (9x - 0.1x), tumour purities (10%-1%) and mutational burdens to provide more confidence in the robustness of our results in Supplementary Figures 1,3 and 4.

>> *Fig 1g poorly described in text*

How was this done? The authors merely state “Using patient data, we next assessed MisMatchFinder’s “Which and how many samples were used? How was dilution done? How do the authors know these signatures are present in the sample in the first place?”

We thank the reviewer for this comment and apologise for the lack of clarity with our previous description. In our revised manuscript’s Figure 1g, three patients are now represented with bladder cancer (estimated tumour purity in plasma of 12%), breast cancer (tumour purity 66%), and melanoma (tumour purity 26%). For the breast and bladder cancer cases, matched tumour/germline sequencing data was available, allowing traditional somatic variant calling

and mutational signature deconvolution to be performed. For the melanoma case, no matching tumour sequencing data was available, so detection of the SBS7a signature was via MisMatchFinder analysis alone. Detailed methods describing how the dilutions and replication were done are now described in the revised manuscript (page 2, lines 67-92 and in Methods page 8).

>> *Concordance is unclear in Fig 1h*

A quick look at the concordance between the stacked diagrams in Fig 1h suggest strong concordance between the tumor/normal and plasma-deepWGS/normal columns (A+B), with lower concordance for their method (column C). This aspect is not highlighted in the text. The authors should quantify the concordance in more detail. Additionally, why are only some signatures highlighted and used in this concordance analysis?

In our revised manuscript, as described above, we now include the original breast cancer case as well as an additional bladder cancer case to examine the concordance between tumour tissue and MisMatchFinder analysis. As suggested by the reviewer, we use the cosine similarity to quantify the concordance of signatures called from high coverage paired tumour/germline data, plasma/germline data, and low-coverage plasma data without a germline control (Figs 1h and 1i). To assess tumour-type specific concordance, we have highlighted those signatures previously found in bladder and breast cancer from large-scale cohorts (Alexandrov et al. 2020⁵). Furthermore, we have now performed the tumour/plasma concordance analysis across varying sequencing depths (9x down to 0.1x) and tumour purities (10% down to 1%) to show that the key mutational signatures remain detectable and consistent even at extreme values to provide further confidence in our results (Supplementary Fig 4). Detailed methods explaining this analysis are now fully described in the revised manuscript (page 6-7, lines 293-317).

>> *Potential for use of tumor/normal samples for additional controls*

The authors could explore if they can use tumor samples to better characterise the limits of the mutational signature detection. Tumor (and their matched normal/germline) samples have the advantage that the ground-truth mutational signatures can be established from the original sample based on accurate mutation calls (tumor vs. normal). The authors could then apply their cfDNA-tailored approach (perhaps without fragment length filter) to explore the detection power when inferring mutational signatures based on single reads in the tumor samples. These single-read results could then be compared to ground truth from standard tumor tissue mutation signature analysis. Furthermore, tumor samples could be downsampled (<10x) and diluted for lower tumor purity (using matched normal) to explore when mutational signatures cannot be reliably detected and quantified. In this testing framework, the authors could explore which types of signatures (e.g. high vs. low mutation load signature) can and cannot be detected with their approach.

We thank the reviewer for this suggestion. Whilst it may be an interesting exercise to explore how our method performs with solid tumour sequencing data, the fragmentomics filter which is critical to the performance of MisMatchFinder, would not be able to be applied in this context as it is specific to ctDNA analysis. As a result, we would not be able to evaluate the performance of our tool on tumour data to characterise the limits of the mutational signature detection. In the revised manuscript we have instead provided further data to validate the performance of MisMatchFinder using additional signatures, further *in silico* analyses and matched tumour/plasma data across different sequencing depths and tumour purities. We believe that these comprehensive additions provide further confidence in the accuracy, generalisability and robustness of our results.

Minor issues

Unsupervised clustering between healthy and cancer samples using SBS signatures revealed APOBEC-linked SBS2 as a major discriminatory signature across most cancer types (Fig. 2b,c).

This sentence appears to be inaccurate. Fig 2b is unsupervised clustering but does not show SBS signatures or how they discriminate between cancer and healthy samples. Fig 2c appears to be a _supervised_ analysis (t-test) comparing SBS2 between cancer and healthy samples”

The reviewer is correct and we apologise for the lack of clarity in this panel. We have now modified the original sentence in the revised manuscript page 3 lines 112-114.

“HRD and PARP inhibitors”. In the introduction the authors state the potential to use PARP inhibitors in patients with HRD signatures. However, it is not clear whether the authors were able to identify HRD signatures in the plasma samples?

In our manuscript, we have used detection of the SBS3 mutational signature in plasma samples as a surrogate for the identification of HRD. SBS3 is strongly associated with germline and somatic *BRCA1* and *BRCA2* mutations as well as *BRCA1* promoter methylation, and has been proposed as a predictor of defective homologous recombination repair and thus the efficacy of therapies exploiting this defect (<https://cancer.sanger.ac.uk/signatures/sbs/sbs3/>).

Issues with external data cohort (EGAS00001003258)

The authors use an external cohort of 273 pan-cancer and 262 healthy LCWGS plasma samples. In the methods section it is stated that this data is obtained from the EGA project with accession EGAS00001003258. However, according to EGA page for this study (<https://ega-archive.org/studies/EGAS00001003258>), this project only contains 118 samples and 236 files. Furthermore, the sample IDs provided in Suppl. Table 3 (EGAFxxx) do not overlap with the sample IDs listed for this EGA project (<https://ega-archive.org/datasets/EGAD00001004939>). This is confusing, and I would encourage the authors to verify the provided data accession codes.

The accession ID previously provided for the external cohort was incorrect and we thank the reviewer for bringing this to our attention. We utilised data from the publication titled “Genome-wide cell-free DNA fragmentation in patients with cancer”¹². The accession ID for this dataset is EGAD00001005339 and this has now been corrected in the revised manuscript.

Data sharing

It is not clear if their internal cohorts used for validation (Fig 1h; Fig 2e,f) will be shared, to enable full transparency and reproducibility of their results.

All internal cohorts used for validation will be shared. The data is currently in the process of being uploaded to EGA and the final EGAD numbers will be provided as soon as this is available. All internal melanoma samples, 39 *BRCA-WT* breast cancer samples and 42 healthy controls are already available in EGA at the following link: <https://ega-archive.org/datasets/EGAD00001011817>

Reviewer #3 (Remarks on code availability):

The software includes instructions on how to run it. However, I have not tested it. It would be great if the authors could provide a small demo file that the software can be tested on.

As per the journal requirements, we have shared the executable binary program of MisMatchFinder along with the source codes and libraries in our bitbucket repository (<https://atlassian.petermac.org.au/bitbucket/projects/SJDAW/repos/mismatchfinder/browse>). The rest of the large files, such as the gnomAD variant information used for the germline filter and the BED file of high mappability regions used as the whitelist for MisMatchfinder, as well as a demo file for MisMatchFinder sanity test, have been uploaded to a Zenodo repository (<https://doi.org/10.5281/zenodo.12754454>). We welcome the reviewer to test this demo file and provide any feedback on the speed and ease of use for our tool.

REFERENCES

1. Wang, N. *et al.* Tool evaluation for the detection of variably sized indels from next generation whole genome and targeted sequencing data. *PLoS Comput Biol* **18**, e1009269 (2022).
2. Wang, N., Khan, S. & Elo, L.L. VarSCAT: A computational tool for sequence context annotations of genomic variants. *PLoS Comput Biol* **19**, e1010727 (2023).
3. Sairam Behera, S.C., Massimiliano Rossi, Sean Truong, Zhuoyi Huang, Michael Ruehle, Arun Visvanath, Gavin Parnaby, Cooper Roddey, Vitor Onuchic, Daniel L Cameron, Adam English, Shyamal Mehtalia, James Han, Rami Mehio, Fritz J Sedlazeck. Comprehensive and accurate genome analysis at scale using DRAGEN accelerated algorithms. *bioRxiv* (2024).
4. Wang, S. *et al.* Copy number signature analysis tool and its application in prostate cancer reveals distinct mutational processes and clinical outcomes. *PLoS Genet* **17**, e1009557 (2021).
5. Alexandrov, L.B. *et al.* The repertoire of mutational signatures in human cancer. *Nature* **578**, 94-101 (2020).
6. Eeckhoutte, A. *et al.* ShallowHRD: detection of homologous recombination deficiency from shallow whole genome sequencing. *Bioinformatics* **36**, 3888-3889 (2020).
7. den Brok, W.D. *et al.* Homologous Recombination Deficiency in Breast Cancer: A Clinical Review. *JCO Precis Oncol* **1**, 1-13 (2017).
8. Kim, Y.S., Lee, M. & Chung, Y.J. Two subtypes of cutaneous melanoma with distinct mutational signatures and clinico-genomic characteristics. *Front Genet* **13**, 987205 (2022).
9. Helen Davies, D.G., Sandro Morganella, Lucy R Yates, Johan Staaf, Xueqing Zou, Manasa Ramakrishna, Sancha Martin, Sandrine Boyault, Anieta M Sieuwerts, Peter T Simpson, Tari A King, Keiran Raine, Jorunn E Eyfjord, Gu Kong, Åke Borg, Ewan Birney, Hendrik G Stunnenberg, Marc J van de Vijver, Anne-Lise Børresen-Dale, John W M Martens, Paul N Span, Sunil R Lakhani, Anne Vincent-Salomon, Christos Sotiriou, Andrew Tutt, Alastair M Thompson, Steven Van Laere, Andrea L Richardson, Alain Viari, Peter J Campbell, Michael R Stratton & Serena Nik-Zainal. HRDetect is a predictor of BRCA1 and BRCA2 deficiency based on mutational signatures. *Nature Medicine*, 517–525 (2017).
10. Zhao, E.Y. *et al.* Homologous Recombination Deficiency and Platinum-Based Therapy Outcomes in Advanced Breast Cancer. *Clin Cancer Res* **23**, 7521-7530 (2017).
11. Andrea Degasperi, T.D.A., Jan Czarnecki, Scott Shooter, Xueqing Zou, Dominik Glodzik, Sandro Morganella, Arjun S. Nanda, Cherif Badja, Gene Koh, Sophie E. Momen, Ilias Georgakopoulos-Soares, João M. L. Dias, Jamie Young, Yasin Memari, Helen Davies & Serena Nik-Zainal. A practical framework and online tool for mutational signature analyses show intertissue variation and driver dependencies. *Nature Cancer* **1**, 249–263 (2020).
12. Cristiano, S. *et al.* Genome-wide cell-free DNA fragmentation in patients with cancer. *Nature* **570**, 385-389 (2019).

RESPONSE TO REVIEWER COMMENTS

We thank the reviewers for their comments on our revised manuscript. We were very pleased with the overall reviewer enthusiasm and recognition of our efforts to address key concerns in the first revision.

We hope to have fully addressed the remaining comments below, as described in our detailed response.

Reviewer #1 (Remarks to the Author):

The authors have addressed all my comments and I believe the manuscript is much improved. I have no further concerns and congratulate the authors for this nice work.

We thank the reviewer for motivating additional work that has significantly strengthened the manuscript.

Reviewer #2 (Remarks to the Author):

I'm happy to report that the authors have done an excellent job addressing all my concerns. I consider all my comments resolved.

We thank the reviewer for their thoughtful suggestions following our initial submission which have made this study and its conclusions stronger.

Reviewer #2 (Remarks on code availability):

I haven't reviewed the code in details, however I can see that a public repository is available, with a README file that explains how to run the code.

Reviewer #3 (Remarks to the Author):

The authors have addressed most of my comments. These two remaining aspects would further increase the quality and rigour of the manuscript, and make it easier for readers and users of the software to understand the assumptions and pitfalls of mutational signature analysis in ctDNA samples:

1) I appreciate the authors new Fig 1f+g exploring how coverage and tumor purity (ctDNA fraction) impacts the recovery of mutational signatures. However, this analysis doesn't explore the (common) worst-case scenario where both of these factors are low (e.g. lpWGS coverage 1x and purity <5%). I would urge the authors to add analysis of more samples with less ideal combinations of coverage and purity in these figures. Please also list the mutational loads (mut/MB) for each of these signatures in their respective samples in the figure.

2) Under limitations and recommendations the authors state that: We recommend that at least 1x coverage for plasma sequencing data is essential for detecting mutational signatures with distinct profiles such as SBS2 and SBS7. However, for non-distinctive signatures such as SBS3, higher sequencing depth (>3x) is necessary to ensure robust results, particularly in samples with <10%.

I recommend that this section also mention that it is more challenging to recover mutational

signatures with low mutational burden (which their results also indicate), and that recovery is impacted by both sequencing coverage, sample purity (ctDNA levels), and tumor type (dictating expected signature load).

Accordingly, the 1x coverage recommendation should ideally be coupled with a recommendation for TP (x%) and minimum signature mutation load (XX mut/mb). We recommend that at least 1x coverage and TP of X% for plasma sequencing of ... such as SBS2 and SBS7 with high mutational loads (XX mut/MB).

We greatly appreciate these comments by the reviewer as well as their constructive critique on our initial submission. We understand why it would be beneficial to investigate “worst-case scenarios” of sequencing coverage and tumour purity combinations to provide users of our tool with a more comprehensive understanding of the limits of detection in our mutational signature analysis framework.

As such, we have included an additional analysis where we have explicitly assessed the joint effect of sequencing coverage and tumour purity on the limit of signature detection for ‘worse-case scenarios’ for these factors (Supplementary Fig.4). Here we assessed detection of SBS2, SBS3 and SBS7a in 20 ctDNA-healthy admixtures from a bladder cancer, a *BRCA*-mutant breast cancer and a melanoma patient respectively. Coverage levels tested span 3x, 2x, 1x, 0.5x and 0.1x while we jointly tested eight tumour purity levels ranging from 1% to 10%.

We have chosen not to annotate the mutational loads (mut/MB) for each of these signatures in the different coverage/purity contexts, as our estimates obtained from MisMatchFinder would not reflect the true tumour mutational burden (TMB) of these samples. MisMatchFinder applies various region-level and fragment-level filters that reduces the effective genome size covered by the sequencing. Along with its read-centric approach to calling mismatches, this would bias the calculation of TMB in our samples and would generally underestimate the true mutational load. Hence, as outlined in the manuscript, we recommend that users sequence healthy controls to similar coverage depths and process this data using the same filters, to derive detection thresholds for high confidence signature calling.

The new analysis provided in Supplementary Fig.4 along with the previous analyses in Figure 1f, g and Supplementary Figure 3, validate the conservative recommendations we provided in the discussion. However, taking into consideration the reviewer’s comments we have expanded our discussion on the limits of detection of our framework as stated below and on Page 4 lines 163-174.

“Overall, our study demonstrates the feasibility of cost-effective ctDNA analysis from LCWGS data for clinically relevant mutational signature detection that can be applied to single samples. We recommend that at least 1x coverage for plasma sequencing data is essential for detecting mutational signatures with distinct profiles such as SBS2 and SBS7 even for estimated tumour purity levels >3%. However, for “flat”, non-distinctive signatures such as SBS3, higher sequencing depth (>3x) is necessary to ensure robust results, particularly in samples with <10% TP. While these are conservative recommendations for recovering mutational signatures in liquid biopsy samples based on our analyses, the limits of detection will also be influenced by the variable mutational burden across different tumour types. Whenever possible, we recommend using a panel of healthy plasma controls of similar sequencing depth and other pre-analytical factors to derive detection thresholds for high confidence ctDNA-based mutational signature signal.”

Reviewer #3 (Remarks on code availability):

The code and software is well documented. I did not test the software.